# Evaluation of wake influence on high-resolution balloon-sonde measurements

Jens Söder[1], Michael Gerding[1], Andreas Schneider[1,a], Andreas Dörnbrack[2], Henrike Wilms[2], Johannes Wagner[2], and Franz-Josef Lübken[1]

[1]Leibniz Institute of Atmospheric Physics at the University of Rostock (IAP), Kühlungsborn, Germany
[2]German Aerospace Center (DLR), Institute of Atmospheric Physics (IPA), Wessling, Germany
[a]now at: SRON Netherlands Institute for Space Research, Utrecht, the Netherlands

*Correspondence to:* Jens Söder (soeder@iap-kborn.de)

**Abstract.** Balloons are used for various in-situ measurements in the atmosphere. On turbulence measurements from rising balloons there is a potential for misinterpreting wake-created fluctuations in the trail of the balloon for atmospheric turbulence. These wake effects have an influence on temperature and humidity measurements from radiosondes as well. The primary aim of this study is to assess the likelihood for wake encounter on the payload below a rising balloon. Therefore, we present a tool for calculating this probability based on radiosonde wind data. This includes a retrieval of vertical winds from the radiosonde and an uncertainty analysis of the wake assessment. Our wake evaluation tool may be used for any balloon-gondola distance and provides a significant refinement compared to existing assessments.

We have analysed wake effects for various balloon-gondola distances applying atmospheric background conditions from a set of 30 radiosondes. For a standard radiosonde we find an average probability for wake encounter of 28 %, pointing out the importance of estimating wake effects on sounding balloons. Furthermore, we find that even millimetre sized objects in the payload can have significant effects on high-resolution turbulence measurements, if they are located upstream of the turbulence sensor.

## 1 Introduction

Since their advent in the beginning of the 20th century (e.g. Aßmann, 1902) rubber-made sounding balloons provide a major platform for atmospheric in-situ soundings of wind, temperature and humidity. To the present day, radiosondes are routinely used for the assimilation of numerical weather predictions (e.g. Bouttier and Kelly, 2001). These balloons are approximately spherically shaped during flight. Horizontally, they drift with the atmospheric wind. Vertically, they rise with a speed of about $5\,\mathrm{m\,s^{-1}}$ relative to the atmosphere. This rise creates a turbulent wake downstream (below) of the balloon (e.g. Taneda, 1978). It mainly depends on the wind-shear in the atmosphere, whether the balloon's wake will hit the sensors on the payload or not. Therefore, great care has to be taken when interpreting turbulence measurements from rising balloons, because they may be influenced by the balloon's wake (Barat et al., 1984). Furthermore, other studies showed an influence of the wake on temperature and humidity measurements from standard radiosondes as well (Tiefenau and Gebbeken, 1989; Gaffen, 1994;

Kräuchi et al., 2016). Wakes can also be generated by other parts of the payload, e.g. ropes holding the gondola. We refer to these objects as the "payload chain".

In the first place, our interest into the subject was triggered because we wanted to improve the data quality on our balloon-borne LITOS instrument (Leibniz-Institute Turbulence Observations in the Stratosphere, Theuerkauf et al., 2011; Schneider et al., 2017). LITOS uses a Constant Temperature Anemometer (CTA) to investigate the Power Spectral Density (PSD) of turbulent velocity fluctuations down to spatial scales of centimetres. This enables us to resolve the transition from the inertial to the viscous subrange of turbulence. We use this approach to retrieve atmospheric energy dissipation rates. Horizontal winds are acquired from a standard radiosonde on the same balloon.

On measurements of turbulent velocity fluctuations, the wake from the balloon can hardly be distinguished from atmospheric turbulence of the same strength. With our LITOS instrument we found that the spectral shape of the velocity fluctuations does not allow a distinction between atmospheric turbulence and wake. Depending on the payload-balloon distance, we found dissipation rates created by the balloon's wake between $10^{-4}\,\mathrm{W\,kg^{-1}}$ and $10^{-2}\,\mathrm{W\,kg^{-1}}$. In terms of aviation turbulence categories, these dissipation rates correspond to "light" and "moderate" turbulence using the scaling of Sharman et al. (2014) for medium-sized aircraft. Wake effects from the ropes holding the gondola show consistently "severe" turbulence intensities around $\sim 10^{-1}\,\mathrm{W\,kg^{-1}}$. Accordingly, these effects should not be neglected for turbulence measurements from rising balloons. For standard radiosondes, Kräuchi et al. (2016) report a warm bias of 1 K on average for a daytime sounding in the stratosphere. Furthermore, moisture from the balloon's skin will lead to a wet bias of stratospheric humidity if the sensor is in the balloon's wake.

Therefore, our question in this study is: Can we determine from a radiosonde measurement, in which altitude the instrument was exposed to turbulence generated from the balloon or from smaller objects in the payload chain?

Pioneering work with regard to the influence of the balloon's wake on turbulence measurements has been done by Barat et al. (1984). They calculated the distance between the wake's centre and the gondola from the wind shear measured with their instrument. They concluded that every altitude bin of their measurement is wake free, where the distance between the wake's centre and the gondola is larger than two balloon diameter. However, they did not consider uncertainties in their wind shear measurement and did not directly include other findings showing that the wake of a spherical body does not have sharp boundaries but is rather fringed at the edges (e.g. Riddhagni et al., 1971). Furthermore, they did not consider vertical winds in their approach. We take those effects into account and refine their technique by applying a probabilistic approach.

The general idea behind our wake evaluation tool is to calculate the advection of the balloon's wake using a radiosonde wind measurement from the same balloon. For every time step, the minimal distance between the wake's centre and the payload is calculated. This concept is similar to the one used by Barat et al. (1984). In contrast however, we take into account the uncertainty in that calculation. Therefore, it is important to consider self-induced motions of the balloon due to changing aerodynamic forces in the critical and supercritical Reynolds number range, because they influence the wind measurement of the radiosonde. Murrow and Henry (1965) conducted tests within a large hangar, whereas MacCready (1965) and Scoggins (1965) examined outdoor launches in still air. A comprehensive review on wind measurements using sounding balloons is

given by Scoggins (1967). We use their results in our study to obtain estimates for the magnitude of those self-induced motions observed on sounding balloons.

Furthermore, we consider that the diameter of the balloon's wake changes on short time scales of a few seconds due to the production of larger vortices. Furthermore, its mean diameter increases on longer timescales in the order of several 10 seconds.

Since the balloon's contour resembles a sphere during flight, we can refer to fundamental experiments done in wind tunnels (e.g. Riddhagni et al., 1971; Gibson and Lin, 1968). An informative visualisation of such a flow can be found in Jang and Lee (2008, Figure 11). More recently, numerical simulations of flows at relevant Reynolds numbers have become available. For example Dommermuth et al. (2002) examined the width of the wake of a sphere in stratified and non-stratified fluids using Large-Eddy-Simulations. They mainly confirm the results of previous laboratory studies.

Additionally, we modify the approach from Barat et al. (1984) by considering vertical winds in our wake evaluation tool. Earlier vertical wind retrievals assumed that all major fluctuations in the ascent rate are comparable to vertical wind fluctuations by gravity waves (e.g. Shutts et al., 1988; Lalas and Einaudi, 1980). Reeder et al. (1999) and Gong and Geller (2010), respectively, subtract a running mean or a second order polynomial from the ascent rate to retrieve vertical winds. Wang et al. (2009) as well as Gallice et al. (2011) extend these approaches by modelling the ascent of the balloon based on a physical

description of the relevant forces. In this work, we present and use a modified version of the Wang et al. (2009) model.

With this new approach, we can calculate the likelihood for encountering the balloon's wake using a radiosonde wind profile. Besides its importance for specially designed turbulence sensors like LITOS, it may be of interest for other studies retrieving turbulent energy dissipation rates from standard radiosondes. The most common method to obtain energy dissipation rates from radiosonde temperature profiles has been adapted from oceanic sciences by Luce et al. (2002) and Clayson and Kantha

(2008). It is frequently referred to as the "Thorpe analysis". Energy dissipation rates are inferred from the vertical displacement (Thorpe displacement) of an air parcel compared to a statically stable profile (Wilson et al., 2010, 2011). Typically, for a standard radiosonde the distance between the balloon and the sensor is between 30 m and 55 m. This makes the measurement susceptible to distortions from the balloon's wake (e.g. Jumper and Murphy, 2001; Kräuchi et al., 2016). Hence, our wake evaluation tool may be used to assess the likelihood of wake influence for every altitude bin of a Thorpe analysis turbulence

retrieval, depending on the balloon-payload distance of the instrument.

Even for longer balloon-payload distances, we cannot expect the balloon's wake to dissolve before it encounters the sensor. Kyrazis et al. (2009) found from a review of laboratory experiments that the wake persists up to 1000 diameters downstream of the balloon.

In the following, we give a short overview on our turbulence evaluation scheme (Section 2). In Section 3, a tool for calculating

the likelihood of wake encounter at the payload position is presented. This includes applying the tool to a set of 30 radiosonde launches. Influences of wake caused by the payload chain are shown in Section 4 and results are discussed in Section 5. In Appendix A the retrieval of vertical winds from a standard radiosonde is presented.

## 2 Retrieving Energy dissipation rates from wind fluctuation data

In order to infer energy dissipation rates, LITOS measures wind fluctuations with two constant temperature anemometers (CTAs) at a frequency of $8\,\text{kHz}$. The handling of these data is explained in this section. Further atmospheric quantities are measured with a radiosonde (Väisälä RS-41: for details please see Survo et al., 2014) and the pendulum motion of the gondola below the balloon is calculated from an inertial sensor (ADIS16407 by Analog Devices, mounted on the LITOS gondola). For measurements in the ascent phase, the LITOS gondola is typically located $180\,\text{m}$ below the balloon (radiosonde: $235\,\text{m}$ below the balloon). Details of the current LITOS instrument have been described by Schneider et al. (2017).

To retrieve atmospheric turbulence we divide our CTA data into time bins of $10\,\text{s}$, calculate the power spectral density (PSD) for each bin and fit the Heisenberg (1948) spectrum of turbulence to the data. In this paper, we will only give a brief overview on retrieving energy dissipation rates, further details can be found in Schneider (2015) and Schneider et al. (2017). For the fit, we use an adaption of the Heisenberg function to velocity fluctuations as a function of angular frequencies $\omega$ given by Schneider (2015, Eq. A.56) based on the idea presented in Lübken (1992):

$$
W(\omega) = C_\text{v}^2 \frac{\Gamma\left(\frac{5}{3}\right)\sin\left(\frac{\pi}{3}\right)}{2\pi w_\text{rel}} \frac{(\omega/w_\text{rel})^{-5/3}}{\left(1 + \left(\frac{\omega}{\omega_0}\right)^{8/3}\right)^2}.
\tag{1}
$$

The structure function constant $C_\text{v}$ and the angular frequency $\omega_0 = \frac{2\pi w_\text{rel}}{l_0}$ (representative of the inner scale $l_0$) are used as fit parameters. $\Gamma$ denotes the gamma function, and $w_\text{rel}$ is the relative vertical velocity between the sensor and the atmosphere. It is calculated from the ascent rate of the balloon $w_\text{asc}$ and the vertical wind $w$:

$$
w_\text{rel} = w_\text{asc} - w.
\tag{2}
$$

Our method for retrieving vertical winds from radiosonde data is explained in Appendix A. Generally, we infer energy dissipation rates $\epsilon$ from the inner scale $l_0$ according to Schneider (2015, Eq. A.48):

$$
\epsilon = c_{l0}^4 \frac{\nu^3}{l_0^4}.
\tag{3}
$$

$c_{l0}$ is a constant depending on the sensor orientation (in our case: $c_{l0} = 15.8028$; Schneider et al., 2017). $\nu$ denotes the kinematic viscosity and is calculated from radiosonde temperatures $T$ and densities $\rho$ (c.f. NOAA, 1976):

$$
\nu = \frac{1.458 \cdot 10^{-6} \cdot T^{3/2}}{\rho(T + 110.4)}.
\tag{4}
$$

Exemplary spectra of velocity fluctuations are given in Figure 1 together with a plot of the fit function (Eq. 1). We find that the fit function generally follows the measured data considering the noise of the PSD. The inner scale $l_0$ is $\mathrm{sim}2.5\,\text{cm}$, underlining the need for high-resolution measurements. The geometric mean of the dissipation rate from both sensors is $9.9\,\text{mW}\,\text{kg}^{-1}$, which corresponds to a moderate turbulence intensity for medium-sized aircraft according to aviation standards (Sharman et al., 2014). The data presented in Figure 1 are taken on a descending balloon. The sensors were located below the gondola, measuring the atmospheric flow unperturbed by any wake effects.

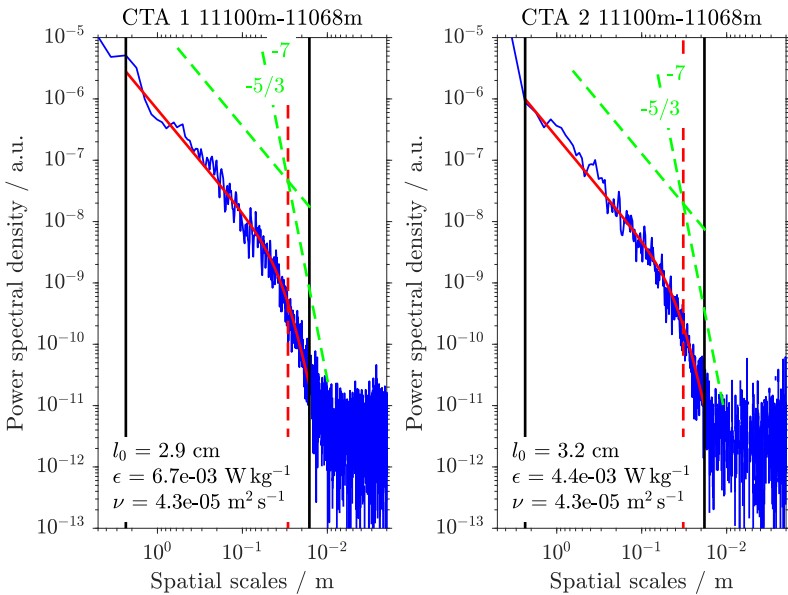

**Figure 1.** Power spectral densities of undisturbed atmospheric turbulent velocity fluctuations (dark blue). The spectra show data from two independent CTA measurements on the same gondola in the same 10 s time span on a descending balloon (06 August 2016). The solid red curve gives the fit of Heisenberg's turbulence model. The transition between the inertial (-5/3) and the viscous (-7) subrange is given by the vertical dashed red line (inner scale $l_0$). Vertical black lines: fit range. Energy dissipation rates $\epsilon$ and kinematic viscosities $\nu$ are given in the lower left corner of each panel.

## 3 Wake caused by the balloon

In this section we describe a "wake evaluation tool" to calculate the likelihood of a wake encounter of the gondola below a rising balloon. Generally, all distances are denoted by a lower-case "d", all diameters by an upper-case "D" and all radii by an upper-case "R". For additional information on the source code please see Section 7. Further below we statistically evaluate a series of 30 radiosonde launches and inspect the influence of wind shear, rotation of the horizontal wind vector, relative vertical velocity and balloon-payload distance by using artificial data.

### 3.1 Method

#### 3.1.1 Payload-wake distance

The concept of our wake evaluation tool is that for every timestep of the calculation, a spherically shaped wake is created from the position of the balloon centre. Each wake is advected with the wind. I.e. the wake does not move, if the fluid is at rest, but is advected horizontally and vertically with the wind. A sketch of the flow within the LITOS payload chain is given in Figure 2. Please note that the depicted path of the wake depends on the wind shear between the balloon and the payload as well as on the

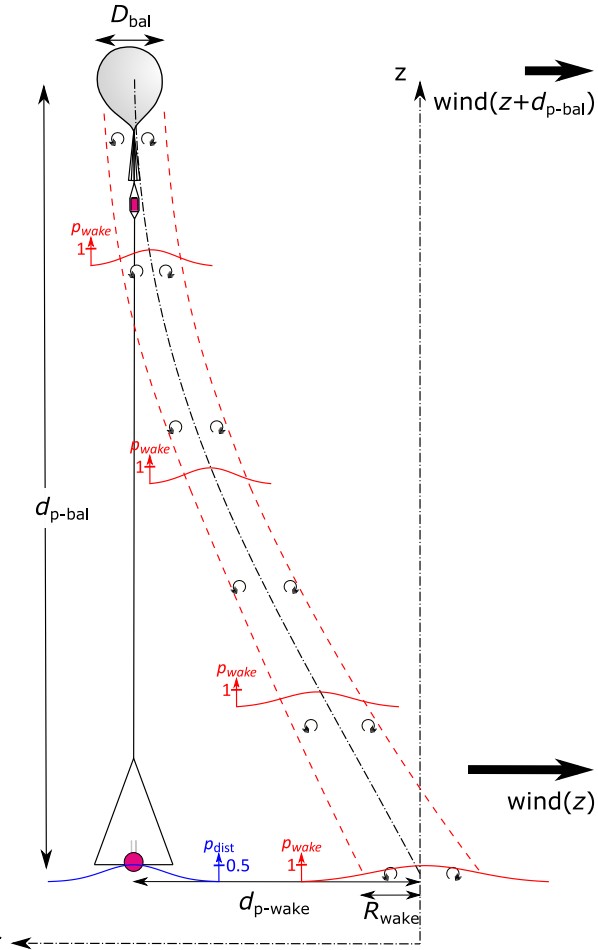

**Figure 2.** Flow within the LITOS payload chain. $d_{p\text{-bal}}$: distance between the payload and the centre of the balloon. $d_{p\text{-wake}}$: distance between the centre of the balloon's wake and the payload. Red lines: distribution of the probability for being in the wake (dashed red line showing the FHWM). Blue line: probability distribution for the payload-wake distance $d_{p\text{-wake}}$. Sketch is not to scale, the radiosonde below the LITOS gondola is omitted for clarity.

relative vertical velocity of the balloon $w_{\text{rel}}$, but not on the magnitude of the wind speed. In order to get the distance between the wake centre and the payload, $d_{p\text{-wake}}$, we look for the closest distance at a specific time between the payload and all created wakes. Due to the intermittent nature of the turbulent wake, its diameter changes on downstream length scales smaller than a few balloon diameters. Therefore, we determine the probability for being in the wake at a certain distance to the wake centre. Furthermore, there is an uncertainty on the calculated payload-wake distance $d_{p\text{-wake}}$ due to measurement errors, which is included in our probabilistic approach. This probability for encountering the balloon's wake at the payload position increases for small balloon-payload distances $d_{p\text{-bal}}$ and low wind shears.

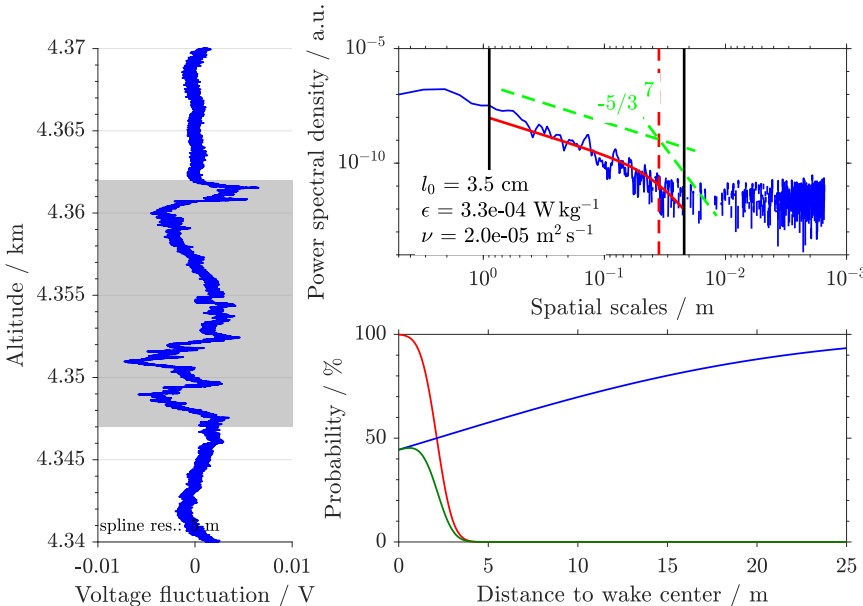

**Figure 3.** Example for turbulence caused by the balloon's wake from a LITOS launch on 29 Jan 2016. Left: Raw data from the CTA, grey shaded area influenced by wake. Top right: Turbulence retrieval fit as in Figure 1 from the data shown in the shaded area. Bottom right: Result from the wake prediction algorithm for the same altitude. Blue: Probability distribution of the payload-wake distance $d_{\text{p-wake}}$ to be in the range of $[0, d]$ after Equation 12. Red: Radial probability distribution of the wake after Equation 13. Green: Combined probability for wake encounter $P_{\text{wake}}$ according to Equation 16 .

The retrieval of $d_{\text{p-wake}}$ is done using radiosonde wind data that have been low-pass filtered with a cut-off frequency of $1/40\,\text{Hz}$ to avoid the influence of self-induced balloon motions on the wind estimate (e.g. MacCready, 1965). For this we use a third order digital Butterworth filter. The position of the payload at the time $t^n = k\tau$ with timestep $\tau$ and $k = 0, 1, 2, ..., n$ may be written as:

$$5 \quad \boldsymbol{X_p}(t^n) = \boldsymbol{X_0} + \sum_{k=0}^{n} \begin{pmatrix} u(t^k) \\ v(t^k) \\ w_{\text{asc}}(t^k) \end{pmatrix} \tau, \tag{5}$$

with $\boldsymbol{X_0}$ being the launch position and $u$, $v$, the horizontal wind components measured by the radiosonde. This equation is valid under the assumption that the combined aerodynamic centre of the balloon and the payload chain is close to the midpoint of the balloon. Therefore, especially for longer payload-balloon distances radiosonde wind data need to be shifted in altitude in order to account for this effect. Furthermore, we expect the payload chain to be hanging straight below the balloon. Any

10 pendulum motions of the gondola are handled as described in Section 3.1.2.

The radiosonde data are given at a rate of $1\,\text{Hz}$. For an average ascent rate of $5\,\text{m s}^{-1}$, this results in a vertical distance of the data points of $5\,\text{m}$, which is in the same order of magnitude as typical payload-wake distances $d_{\text{p-wake}}$ especially for short payload-balloon distances $d_{\text{p-bal}}$ below $100\,\text{m}$. In order to avoid large errors on the payload-wake distance due to this coarse

gridding, we linearly interpolate the radiosonde data to a timestep $\tau = 1/5\,\mathrm{s}$.

For each time $t^n$, there will be a wake created at the position of the balloon $\boldsymbol{X}_{\mathbf{bal}}(t^n)$, which is located $d_{\text{p-bal}}$ above $\boldsymbol{X}_{\boldsymbol{p}}(t^n)$. We assume the position of the centre of that particular wake $\boldsymbol{X}_{\mathbf{wake}}(t^m)$ to be moving with the wind taken at the previous balloon position at time $t^n$. This means that the balloon's wake moves with the background wind that is measured at time $t^n$. In other words, we assume this background wind to be constant for the time it takes to lift the payload through the payload-balloon distance:

$$
\boldsymbol{X}_{\mathbf{wake}}(t^l) = \boldsymbol{X}_{\mathbf{bal}}(t^n) + m \begin{pmatrix} u(t^n) \\ v(t^n) \\ w(t^n) \end{pmatrix} \tau,
\tag{6}
$$

with $t^l = t^n + m\tau$, $m = 0, 1, 2, ..., L$ and $L = \left( \frac{C_{\text{sh}}\, d_{\text{p-bal}}}{\min(w_{\text{rel}})\tau} \right)$. In this case, $\min(w_{\text{rel}})$ denotes the minimal relative ascent rate during the whole flight (c.f. Appendix A). $L$ is an estimate for the maximum number of timesteps the payload needs to fly through the payload-balloon distance. It is introduced in order to save computational power and does not have a physical meaning in terms of the wake's life time. The constant $C_{\text{sh}} = 1.2$ accounts for a possible change in wind shear that would increase the number of timesteps until the wake reaches its closest position to the payload. For each time $t^l$, we calculate the minimal Euclidian distance between the LITOS gondola and the wake centre:

$$
d_{\text{p-wake}}(t^l) = \min_{m=0,...,L} |\boldsymbol{X}_{\boldsymbol{p}}(t^l) - \boldsymbol{X}_{\mathbf{wake}}(t^l)|.
\tag{7}
$$

An example for an altitude range of our LITOS data affected by the balloon's wake is shown in Figure 3.

### 3.1.2 Uncertainty in the payload-wake distance

Due to measurement errors of the radiosonde and self-induced motions of the balloon caused by aerodynamic forces, there is an uncertainty on the payload-wake distance $d_{\text{p-wake}}$. It will be governed by three main components: the uncertainty in the wind measurement $\Delta_U$, the uncertainty in the position of the payload $\Delta_{X_\text{p}}$, and the uncertainty in the position of the balloon. $\Delta_U$ consists of the uncertainty of the vertical wind retrieval and the uncertainty of the radiosonde measurement (resulting from the uncertainty in the GPS position). The error in the vertical wind $\Delta_w$ is assumed to be $2\,\mathrm{m\,s^{-1}}$ below $19.5\,\mathrm{km}$ altitude and $1\,\mathrm{m\,s^{-1}}$ above (c.f. Appendix A). Below $19.5\,\mathrm{km}$ however, we set $w = 0$. We assume the shear of the vertical wind between the balloon and the gondola to be no larger than $1\,\mathrm{m\,s^{-1}}/d_{\text{p-wake}}$. Therefore, the uncertainty in the vertical wind measurement relevant for our wake evaluation is $\Delta_w = 1\,\mathrm{m\,s^{-1}}$. After Vaisala (2018), there is a measurement uncertainty of $\Delta_{\text{RS}} = 0.15\,\mathrm{m\,s^{-1}}$ in the horizontal wind speed. Generally, self-induced motions of the balloon would add to the uncertainty in the wind measurement as well. However, the period of these self-induced balloon motions is found to be typically sufficiently below $40\,\mathrm{s}$. Due to the low pass filtering of the wind data (3rd order Butterworth, $1/40\,\mathrm{Hz}$ cut-off frequency), the effect of these motions on the wind measurement is negligible and does not need to be considered in the error estimate. Therefore, we assume for the

uncertainty in the horizontal wind:

$$
\mathbf{\Delta}_U = \begin{pmatrix} \frac{\Delta_{\mathrm{RS}}}{\sqrt{2}} \\ \frac{\Delta_{\mathrm{RS}}}{\sqrt{2}} \\ \Delta_w \end{pmatrix} \tag{8}
$$

In case of a LITOS launch, the uncertainty in the position of the payload $\mathbf{\Delta}_{X_\mathrm{p}}$ is acquired from the motion sensor on board, from which a horizontal payload displacement $\Delta_{\mathrm{P_{horz}}}$ is taken. As there is no information on the direction of the displacement available, we assume it to be equally distributed. The error on the vertical payload position is given by the vertical grid step of the radiosonde after interpolation as described in Section 3.1.1. The vertical payload displacement can be neglected because it is attached to the balloon by a string.

$$
\mathbf{\Delta}_{X_\mathrm{p}} = \begin{pmatrix} \frac{\Delta_{\mathrm{P_{horz}}}}{\sqrt{2}} \\ \frac{\Delta_{\mathrm{P_{horz}}}}{\sqrt{2}} \\ w_{\mathrm{asc}}\tau \end{pmatrix} \tag{9}
$$

As discussed above, the balloon is subject to self-induced horizontal motions due to aerodynamic forces in the critical and super critical Reynolds number range. They affect the balloon position at the time of wake creation. MacCready (1965) estimates the maximum amplitude of these horizontal motions to be $\Delta_{X_{\mathrm{bal_{horz}}}} = 2.8 D_{\mathrm{bal}}(1+2m_{\mathrm{r}})^{-1}$, with $D_{\mathrm{bal}}$ denoting the balloon diameter and $m_{\mathrm{r}}$ the relative mass of the sphere to the displaced air. Typically, $D_{\mathrm{bal}}$ is below $10\,\mathrm{m}$. In the case of the LITOS launch from 29 January 2016 (discussed in Section 3.1.4) the mean amplitude in the critical and supercrititcal Reynolds number range is $6.3\,\mathrm{m}$. The error in the vertical balloon position is given by the vertical grid step of the radiosonde after interpolation:

$$
\mathbf{\Delta}_{X_\mathrm{bal}} = w_{\mathrm{asc}}\tau \begin{pmatrix} 0 \\ 0 \\ 1 \end{pmatrix} + \frac{1}{\sqrt{2}} \begin{pmatrix} 1 \\ 1 \\ 0 \end{pmatrix}
$$

$$
\cdot \begin{cases} 2.8 D_{\mathrm{bal}}(1+2m_{\mathrm{r}})^{-1} & \text{if } Re \geq 2\cdot10^5 \\ 0 & \text{if } Re < 2\cdot10^5. \end{cases} \tag{10}
$$

In order to estimate the uncertainty in the payload-wake distance $d_{\mathrm{p\text{-}wake}}$, we use a first order Taylor series expansion. Assuming independent variables, we obtain for the uncertainty of the payload-wake distance $\Delta_{d_{\mathrm{p\text{-}wake}}}$:

$$
\Delta_{d_{\mathrm{p\text{-}wake}}} = \sqrt{(m\tau\mathbf{\Delta}_U)^2 + \mathbf{\Delta}^2_{X_\mathrm{p}} + \mathbf{\Delta}^2_{X_\mathrm{bal}}} \tag{11}
$$

$m$ denotes the number of timesteps between the creation of the wake and its closest encounter with the payload (c.f. Section 3.1.1).

### 3.1.3 Probability of wake encounter

Due to the uncertainty in the payload-wake distance $\Delta_{d_{\mathrm{p\text{-}wake}}}$, we take a probabilistic approach to asses whether our instrument was affected by the balloon's wake. We assume the probability distribution of payload-wake distances to be Gaussian shaped.

In order to assess the probability for wake encounter, we first calculate for every radial distance $r$ between the payload and the wake centre the probability $\Phi$ that the true payload-wake distance $d_{\text{p-wake}}$ is smaller than $r$. $\Phi$ is given by a cumulative Gaussian distribution with a standard deviation of $\frac{\Delta_{d_{\text{p-wake}}}}{2}$ and a mean of $d_{\text{p-wake}}$ (blue curve in the right panel of Figure 3):

$$\Phi(r \mid d_{\text{p-wake}}, \Delta_{d_{\text{p-wake}}}) = \frac{\sqrt{2}}{\Delta_{d_{\text{p-wake}}} \sqrt{\pi}} \int_{-\infty}^{r} e^{-\frac{(y - d_{\text{p-wake}})^2}{\Delta_{d_{\text{p-wake}}}}} \, dy \tag{12}$$

According to Barat et al. (1984), all measurements where $d_{\text{p-wake}}$ is smaller than two balloon diameters are likely to be influenced by the balloon's wake, regardless of the balloon-gondola distance. Numerical experiments using Detached Eddy Simulation from Constantinescu and Squires (2004) however show that the width of the turbulent wake of a sphere at the relevant Reynolds numbers ($Re = 50{,}000...850{,}000$) depends on the distance to the sphere. Riddhagni et al. (1971) showed from wind tunnel measurements that the radial distribution of the probability for being in the wake is given by a cumulative

Gaussian distribution (c.f. Figure 2). Therefore, we calculate the probability $\Psi$ of being in the wake for any radial distance $r$ to the wake centre by (red curve in the right panel of Figure 3):

$$\Psi(r \mid R_{\text{wake}}) = 1 - \frac{3}{R_{\text{wake}} \sqrt{2\pi}} \int_{-\infty}^{r} e^{-\frac{3(y - R_{\text{wake}})^2}{2 R_{\text{wake}}}} \, dy \tag{13}$$

$R_{\text{wake}}$ denotes the radius of the wake. According to Riddhagni et al. (1971) $R_{\text{wake}}$ is the mean of the distribution $\Psi$ and $R_{\text{wake}}/3$ its standard deviation. Namely, the radial distance to the wake centre with a probability for being in the wake of $50\,\%$

(FWHM). It is shown by a dashed red line in Figure 2. In z-direction, $R_{\text{wake}}$ is approximately constant up to six diameters downstream of the sphere and grows for larger distances. This growth of the wake radius is written as (c.f. Riddhagni et al., 1971):

$$R_{\text{wake}} = A \cdot \begin{cases} \left( \frac{d_{\text{p-bal}}}{D_{\text{bal}}} \right)^{\frac{1}{3}} & \text{if } d_{\text{p-bal}} \geq 6 \, D_{\text{bal}} \\ 1 & \text{if } d_{\text{p-bal}} < 6 \, D_{\text{bal}}. \end{cases} \tag{14}$$

$d_{\text{p-bal}}$ is the distance between the balloon and the payload and $D_{\text{bal}}$ the balloon diameter. For the constant $A$, Riddhagni et al.

(1971) gave $A = 0.7$, whereas Dommermuth et al. (2002) found $A = 0.5$ from Large Eddy Simulations. We choose $A = 0.7$ for our calculations to avoid underestimating the wake diameter.

We consider both distributions $\Phi$ and $\Psi$ as independent of each other, because Equation 12 describes the uncertainty of the calculated payload-wake distance, whereas Equation 13 describes the intermittency of the wake. Therefore, the joint probability

for a wake encounter as a function of the distance to the wake centre $d$ is given by the product of both distributions (green line in the bottom right panel of Figure 3):

$$P_{\text{wake}}(d \mid R_{\text{wake}}, d_{\text{p-wake}}, \Delta_{d_{\text{p-wake}}}) =$$
$$\Phi(d \mid d_{\text{p-wake}}, \Delta_{d_{\text{p-wake}}}) \cdot \Psi(d \mid R_{\text{wake}}) \tag{15}$$

The most likely distance $d$ between the payload and the wake centre is given by the maximum in Eq. 15 (maximum of the green line in the right panel of Figure 3). Therefore, the probability for wake encounter at a given altitude is written as:

$$P_{\text{wake}}(R_{\text{wake}}, d_{\text{p-wake}}, \Delta_{d_{\text{p-wake}}}) =$$
$$\max\left(\Phi(d \,|\, d_{\text{p-wake}}, \Delta_{d_{\text{p-wake}}}) \cdot \Psi(d \,|\, R_{\text{wake}})\right) \tag{16}$$

Generally, we consider every datapoint as wake-free, where the probability for wake encounter is below $P_{\text{wake}} = 5\,\%$. An example for a wake influenced data section can be seen in Figure 3. The raw velocity fluctuation data (spline subtracted) show a relatively narrow turbulent patch of $15\,\text{m}$ with no transition from turbulent to non-turbulent regions. The wake probability in this region is comparatively high ($47\,\%$). In the spectrum of the data, we notice no clear transition from the -5/3 to the -7 range. This may be due to the inhomogeneity in the turbulence field, because the balloon does not continuously stay in the centre of

the wake. The retrieved energy dissipation rate is $0.3\,\text{mW}\,\text{kg}^{-1}$. This is about one and a half orders of magnitude lower than the true atmospheric turbulence shown in Figure 1. We like to stress however that the data underlying Figure 1 show only one exemplary case of a turbulent altitude bin. Under different atmospheric conditions, LITOS measured atmospheric turbulence patches ranging from $0.001\,\text{mW}\,\text{kg}^{-1}$ to $100\,\text{mW}\,\text{kg}^{-1}$. In contrast, we typically find dissipation between $0.01\,\text{mW}\,\text{kg}^{-1}$ and $1\,\text{mW}\,\text{kg}^{-1}$ for balloon-wake induced turbulence. However, during previous measurements with lower payload-balloon

distances we measured wake induced turbulence stronger than $10\,\text{mW}\,\text{kg}^{-1}$ (data not shown here).

### 3.1.4  Exemplary wake-encounter probabilities for 29 Jan 2016

The payload-wake distance $d_{\text{p-wake}}$ for the LITOS launch on 29 January 2016 is shown as a solid green line in the right panel of Figure 4. Evaluating the whole LITOS flight, we find for none of the altitude bins a wake probability of more than $95\,\%$ (considered as wake affected). On the other hand, $69.1\,\%$ of all altitude bins are considered wake free ($P_{\text{wake}} < 5\,\%$), whereas

the mean probability for a wake encounter over the whole flight is $5.6\,\%$. According to the criterion by Barat et al. (1984), $3.5\,\%$ of all altitude bins are affected by the balloon's wake. The percentage of truly turbulent altitude bins (turbulence detected without any wake influence) for this flight however, is $6.0\,\%$. Therefore, the occurrence rate of wake is in the same order of magnitude as the occurrence rate of atmospheric turbulence. This underlines the importance of wake identification analysis, as the wake adds a considerable amount of false turbulence detections.

### 3.2  Statistical evaluation of wake encounter probability

### 3.2.1  Influence of the payload-balloon distance for realistic soundings

We use a series of Vaisala RS41 radiosondes to evaluate typical percentages of wake influence as a function of the payload-balloon distance $d_{\text{p-bal}}$ for arbitrary payloads. This will allow users of specialised payloads to asses their risk of wake encounter. The data set has been acquired at Kiruna in Northern Sweden during the GW-LCycle II campaign in January/February 2016. In

order to consistently retrieve vertical winds, we take only those sondes into account where the uplift during the filling process was measured and a $500\,\text{g}$ balloon has been used (30 in total). First, the vertical wind during each flight is calculated using

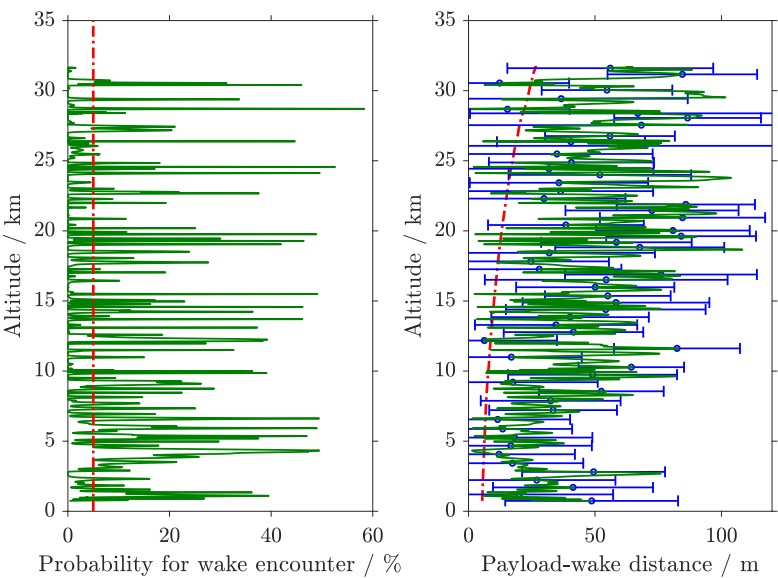

**Figure 4.** Wake assessment for LITOS flight on 29 Jan 2016 ($d_{\text{p-bal}} = 180\,\text{m}$). Left: Probability for wake encounter on the LITOS payload (green). Probabilities smaller than $5\,\%$ are considered to be wake-free (red line). Right: Distance between the centre of the wake and the gondola ($d_{\text{p-wake}}$, green line) with errorbars (blue). Upper limit for wake-free data according to Barat et al. (1984) in red. For clarity, only every $100^{\text{th}}$ errorbar (blue) is shown.

the approach presented in Appendix A. Second, the likelihood for wake encounter is computed for every altitude bin of every radiosonde according to Section 3.1. This calculation has been done for payload-balloon distances $d_{\text{p-bal}}$ between $20\,\text{m}$ and $200\,\text{m}$ with a spacing of $10\,\text{m}$. This is possible, because instead of measuring it directly, our wake prediction algorithm calcu-lates the wind shear between the balloon and the payload and can therefore simulate any payload-balloon distance. In contrast

to the LITOS-payload, on radiosondes the angle by which the payload is displaced from the vertical is not measured. From several LITOS-flights with different payload-balloon distances we know however that the typical standard deviation of the displacement angle is around $1°$, regardless of the payload-balloon distance. As the weight to cross-sectional area ratio of the LITOS-payload and the RS41 radiosonde is similar ($9.8\,\text{kg}\,\text{m}^{-2}$ and $8.7\,\text{kg}\,\text{m}^{-2}$, respectively), we assume their pendulum am-plitude to be comparable, even though their shape is different. Accordingly, we use the standard deviation of the displacement

angle from the LITOS measurement presented in Section 3.1.4 ($0.93°$). It is used for the uncertainty propagation as described in Section 3.1.2.

In Figure 5 the probabilities for wake encounter are shown. These probabilities are averaged over all altitude bins and all flights. We notice that for a balloon-gondola distance of $30\,\text{m}$ (older radiosondes), nearly $100\,\%$ of the altitude bins are poten-tially wake affected ($P_{\text{wake}} > 5\,\%$) and the mean probability for wake encounter at the position of the gondola is $40\,\%$. For a

$55\,\text{m}$ distance (currently used by Vaisala), we find approximately $4\,\%$ of all altitude bins to be certainly free of wake influence and a mean probability for wake encounter of about $28\,\%$. For larger distances, the percentage of potentially wake influenced

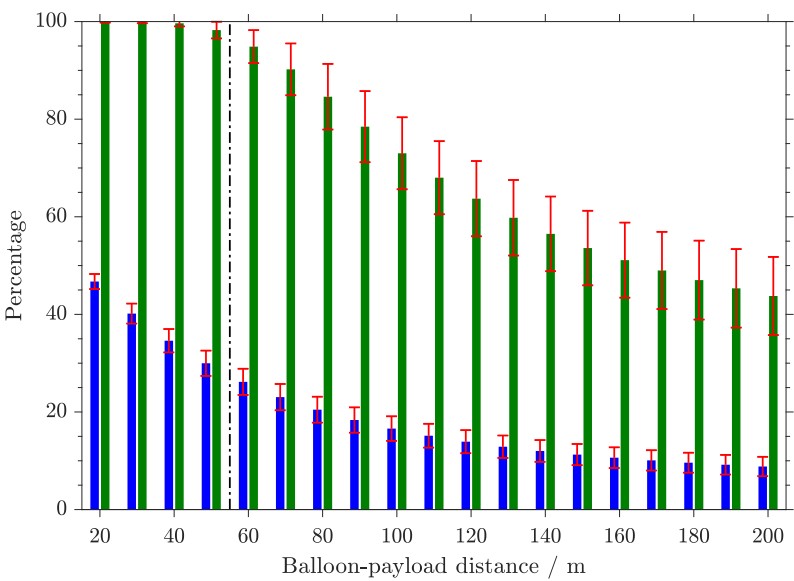

**Figure 5.** Blue: Mean probability of wake encounter using background information from 30 radiosondes flights. Green: Percentage of potentially wake affected altitude bins ($P_{\mathrm{wake}} > 5\,\%$). Red: Standard deviation between different launches. Black: payload-balloon distance of a standard radiosonde.

altitude bins decreases, reaching about $44\,\%$ for a distance of $200\,\mathrm{m}$ with an average wake probability of $8.8\,\%$.

### 3.2.2 Influence of the payload-balloon distance and other parameters for idealised soundings

In this section, we demonstrate the quantitative influence of wind shear (Figure 6), rotation of the horizontal wind vector (Figure 7) and relative vertical balloon velocity $w_{\mathrm{rel}}$ (Figure 8) on the payload-wake distance $d_{\mathrm{p\text{-}wake}}$ for different payload-balloon distances $d_{\mathrm{p\text{-}bal}}$. For that we apply the software described in Section 3.1 on an artificial dataset with constant shear, rotation and $w_{\mathrm{rel}}$, where each of these three parameters is individually and systematically changed. This dataset is based on a typical radiosonde with a $500\,\mathrm{g}$ rubber balloon, in line with the real data used in Section 3.2.1. In order to separate the influence of the three parameters on the wake encounter probability from instrumental effects, we did these calculations assuming an idealized instrument with no self-induced balloon motions, no pendulum motions and no measurement uncertainties of the radiosonde. Furthermore, histograms showing the frequency distribution of the above parameters are obtained from the mentioned radiosonde dataset and used to demonstrate typical values. For these histograms the respective parameter (e.g. wind shear) is calculated as a mean over the standard balloon-radiosonde distance of $55\,\mathrm{m}$.

In Figure 6 we see that the risk of being in the wake increases with decreasing wind shear $dU/dz$ and decreasing balloon-gondola distance, as expected. In this case we have assumed no rotation of the wind vector and a relative vertical balloon

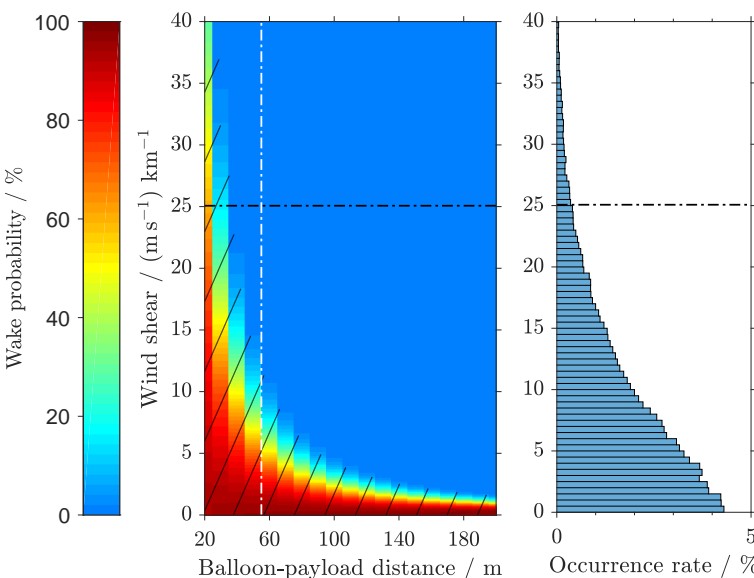

**Figure 6.** Wind shear influence on the wake probability $P_{\mathrm{wake}}$ (no wind rotation, relative vertical balloon velocity $w_{\mathrm{rel}}$: $5\,\mathrm{m\,s}^{-1}$). The ninety-fifth percentile of the measured wind shear is indicated by the black dashed-dotted line. Left: Wake probability as a function of the magnitude of the wind shear and payload-balloon distance $d_{\mathrm{p\text{-}bal}}$. The white line denotes the horizontal balloon gondola distance for the RS41 radiosonde, the hatched area is potentially affected by the balloon's wake ($P_{\mathrm{wake}} > 5\,\%$). Right: Wind shear from the 30 radiosonde observations described in Section 3.2.1.

velocity of $w_{\mathrm{rel}} = 5\,\mathrm{m\,s}^{-1}$. The hatched area denotes cases with a probability for wake encounter of more than $5\,\%$. From the plot we see that for a standard radiosonde configuration ($d_{\mathrm{p\text{-}bal}} = 55\,\mathrm{m}$) we need a wind shear of more than $12\,\mathrm{m\,s}^{-1}\,\mathrm{km}^{-1}$ to achieve a wake probability of less than $5\,\%$. The statistical distribution of the measurements in the right panel shows the occurrence rate of a certain shear over the payload-balloon distance of a standard radiosonde ($55\,\mathrm{m}$). As can be seen, these

higher wind shears occur for about $30\,\%$ of all altitude bins.

Similarly, according to Figure 7 the risk of a wake encounter is reduced, if there is a rotation in the horizontal wind vector. The effect of a wind rotation on the payload-wake distance however, depends on the wind speed. This is, because a rotation of the wind vector leads to a larger separation of the wake and the payload for stronger wind speeds. Therefore, we plotted the wake probability as a function of the payload-balloon distance and the rotation of the horizontal wind vector multiplied by the wind

speed. The dataset is created without wind shear, with a relative vertical balloon velocity of $w_{\mathrm{rel}} = 5\,\mathrm{m\,s}^{-1}$ and a typical wind speed of $|\boldsymbol{U}| = 20\,\mathrm{m\,s}^{-1}$. For the occurrence rate in the right panel it should be noted that the resolution of the wind direction measurement by the radiosonde is only $1°$. Combined with the averaging over the $55\,\mathrm{m}$ balloon-radiosonde distance, this leads to the exceptionally high occurrence rate in the lowest bin of the right panel.

Another parameter influencing the probability for wake encounter is the relative vertical velocity between the balloon and the

atmosphere $w_{\mathrm{rel}}$. As expected, we find a higher probability for wake encounter for higher relative velocities (c.f. Figure 8). For

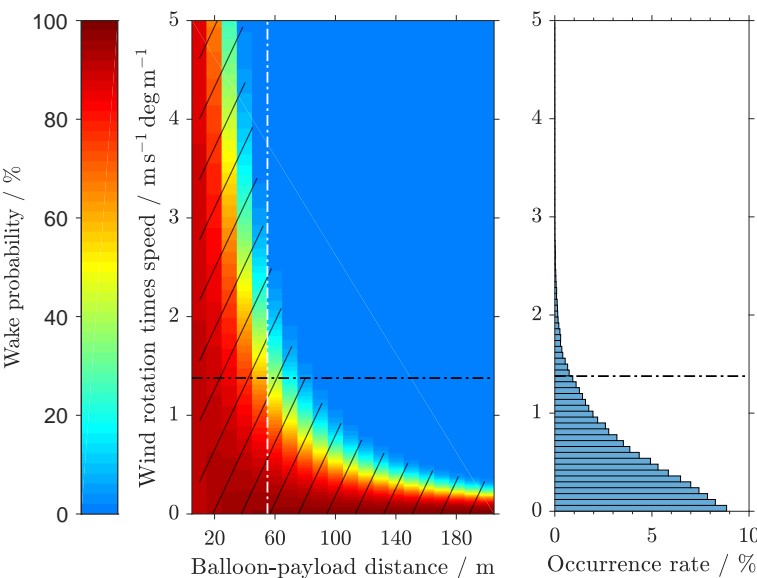

**Figure 7.** Same as Figure 6, but showing the influence of wind rotation on the wake probability (no wind shear, relative vertical balloon velocity $w_{\mathrm{rel}}$: $5\,\mathrm{m\,s^{-1}}$).

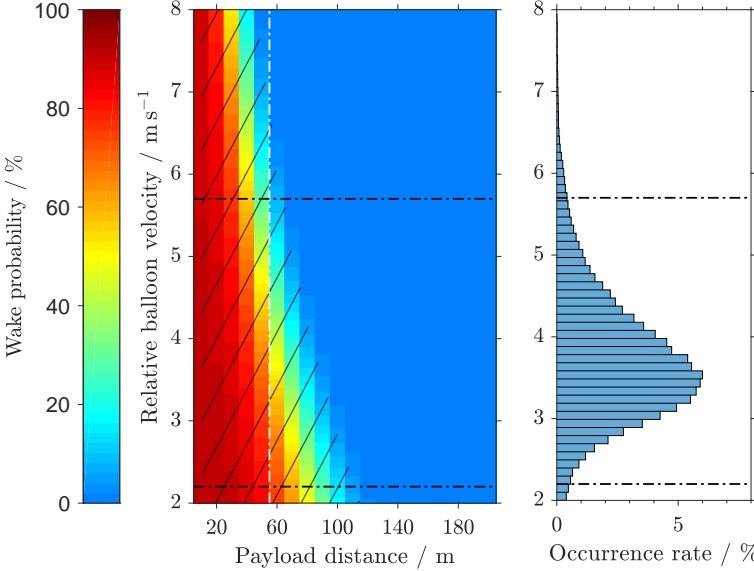

**Figure 8.** Same as Figure 6, but showing the influence of the relative vertical balloon velocity $w_{\mathrm{rel}}$ on the payload-wake distance (wind shear: $10\,\mathrm{m\,s^{-1}\,km^{-1}}$, no wind rotation).

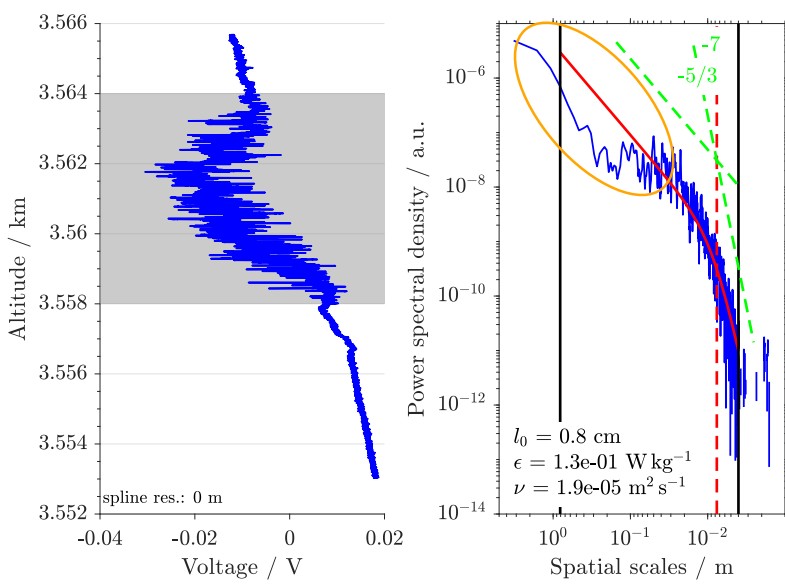

**Figure 9.** Wake of a rope holding the gondola. Left: Raw data from the CTA. Right: Turbulence retrieval fit as in Figure 1 from the data shown in the left panel. The orange ellipse denotes a region of reduced power spectral densities compared to the Heisenberg model (Eq. 1). This points to turbulence in the wake of a small object in the vicinity of the sensor.

our radiosonde dataset the peak of the velocity distribution is at $w_{\mathrm{rel}} = 3.5\,\mathrm{m\,s^{-1}}$.

From the analysis of these three parameters, we find that within the ninety-fifth percentile of the wind shear for the given radiosonde dataset, the wake probability changes from 0.1 % to 96 %. Within the ninety-fifth percentile of all examined rotations in the horizontal wind vector the wake probability changes from 68 % to 95 %. For the variation in relative vertical balloon ve-

locity the wake probability changes from 43 % to 84 %. Therefore, we conclude that within the spread of the given radiosonde dataset wind shear has the strongest influence on the likelihood for wake encounter.

## 4   Wake caused by the payload chain

Another possible cause for self-induced turbulence are smaller objects of sub-metre size in the payload chain above the sensor. In the case of the LITOS instrument, these are mainly the ropes holding the gondola. Both turbulence sensors used on this

instrument are placed above the gondola. In the following, this will be used to exemplarily describe the effect of such an object on a high-resolution turbulence measurement.

Under unfavourable conditions, the wind shear is such that the wake of the ropes is advected to the sensors which are located below the ropes at a minimum downstream distance of 15 cm. At a rope diameter of 1 mm, the Reynolds number of the flow is between $Re = 400$ on ground level and $Re = 5$ at 32 km. For $Re > 50$ (occurring in altitudes below ~19.5 km), a Kármán

vortex street forms in the flow downstream of a cylinder (Williamson, 1996; Henderson, 1995). We do not see the vortex shed-

ding frequency in our data, which is expected because Kármán vortex streets completely break down into turbulence in the far field of the flow more than 50 diameters away from the source (Roshko, 1954; Taneda, 1959). We expect this breakdown of the vortex street to be enhanced by the surface roughness of the rope. Consistently, strong turbulent velocity fluctuations that show a different spectral shape compared to atmospheric turbulence are seen on the CTA data if one of the sensors is hit by such a

collapsed vortex street (c.f. Figure 9). For $Re < 50$, we notice a deceleration of the flow if the sensor gets into the trail of one of the ropes but no fine-scale turbulent fluctuations. This results in a different spectral shape than the one presented in Figure 9. The difference between rope-wake related and atmospheric turbulence manifests in a drop of power spectral densities at scales above 10 cm compared to the fit function (Eq. 1). It is explained by the measurement geometry: assuming isotropy of the turbulent flow the largest eddies can not be larger than the distance from the wake source to the sensor (15 cm here), because

otherwise the growth speed of the eddies would be larger than the speed of the flow around the object. Another property of these rope-wake induced turbulence is its strong local confinement: The turbulent region shown in Figure 9 has a length of only six metres along the flight track and shows a comparatively high energy dissipation rate of $130\,\mathrm{mW\,kg^{-1}}$ with hardly any transition between turbulent and non-turbulent regions.

In order to discard artificial turbulence of this kind in our LITOS retrieval we inspect every spectrum of turbulent velocity fluctuations together with its raw data by eye and discard all altitude bins that show the above mentioned distortions of the measured spectrum compared to the fit function. An automated detection does not seem feasible to us, as the changes in the spectral shape can be very subtle (presumably depending on the relative speed of the gondola $w_{\mathrm{rel}}$) and are therefore difficult to capture by a criterion like the mean squared distance between the data and the fit function.

## 5   Discussion

Our method of measuring turbulent energy dissipation rates relies on resolving the inner scale of turbulence. Other measuring techniques determine the energy dissipation rate from the structure function or from the power spectral density $W(k) = A\varepsilon^{2/3}k^{-5/3}$ of the horizontal wind in the inertial subrange (Barat, 1982a, $A$ denotes a constant). The advantage of the latter methods is that the power spectrum does not need to resolve the inner scale $l_0$ but can be measured at larger spatial scales. Therefore, the

magnitude of the fluctuations is higher, which makes them easier to measure (c.f. Figure 1). Consequently, our method is more susceptible to wake related influences or other technical distortions due to the lower fluctuation amplitudes. On the other hand, their method requires a calibration of the anemometer that forced them to use comparatively complicated ionic anemometers (e.g. Barat, 1982b). Furthermore, our technique is more suitable on sounding balloons, because the balloon diameter (between 2 m and 13 m, depending on altitude) is considerably larger than the fluctuation scales we are evaluating (order of a few cen-

timetres). Therefore, our method avoids a filtering of measured fluctuations due to balloon movements, as reported by Barat et al. (1984). Additionally, their method depends on the assessment of parameters like the Richardson number in the turbulent layer that cannot be measured directly (Barat and Bertin, 1984a).

Considering vertical winds for the LITOS retrieval and for the payload-wake distance calculation is beneficial, because the LITOS balloon from 29 Jan 2016 as well as the radiosonde series reached ascent rate variations in the mid stratosphere of $\pm 5\,\mathrm{m\,s^{-1}}$. Assuming the balloon to show a constant relative vertical velocity $w_\mathrm{rel}$ (i.e. subtracting a mean ascent rate to obtain the vertical wind) would result in errors of up to $\pm 2.5\,\mathrm{m\,s^{-1}}$ in $w_\mathrm{rel}$ (data not shown), compared to $\pm 1\,\mathrm{m\,s^{-1}}$ from our retrieval. Unfortunately, to our knowledge, there is no model that describes the strong variations in the vertical velocity of a balloon relative to the background vertical velocity for Reynold numbers in the critical and supercritical range. Correspondingly, a measurement of $w_\mathrm{rel}$ in the troposphere and lower stratosphere on the LITOS system using either a sonic anemometer or a Pitot tube is highly desirable.

Concerned with the significance of wake related issues for our LITOS measurements we found that turbulence induced by the balloon as well as by the ropes has substantial effects on the raw data. When comparing turbulence measurements from rising balloons to a first measurement on a descending balloon (exemplary velocity spectrum shown in Figure 1), we found several turbulent areas of only a few ten meters in altitude on the ascent data. They are suspicious for balloon-wake influence. Hence we developed the model for the propagation of the balloon-wake based on radiosonde wind data described in this paper. With respect to the verification of our wake detection algorithm we found that there is no altitude bin during the LITOS flight from 29 Jan 2016 where our model predicts a certain wake encounter (wake probability higher than 95 %). This is due to the relatively large balloon gondola distance. It does not mean that no wake encounter took place. Instead, the low abundance of regions with high wake probability reflects the level of uncertainty in the calculation. Furthermore, the measured dissipation rates of balloon induced turbulence are lower than the ones created by the wake of small objects in the payload chain ($10^{-4}\,\mathrm{W\,kg^{-1}}$ vs. $10^{-1}\,\mathrm{W\,kg^{-1}}$) due to the exponential decay of turbulent intensity with time (Gibson and Lin, 1968). This means that whenever both wakes occur at the same time, the balloon's wake cannot be detected, because it is masked by the stronger wake of the ropes holding the gondola. This is the case for about one third of the dataset. When trying to identify balloon-wake related turbulence by its spectral shape, we found that it shows a less distinct transition from the inertial to the viscous subrange of the spectrum compared to atmospheric turbulence (c.f. Figure 3). This however resembles atmospheric turbulence in case of a low signal to noise ratio (occurring for low dissipation rates of $< 10^{-4}\,\mathrm{W\,kg^{-1}}$ and for low pressures of $< 100\,\mathrm{hPa}$) and therefore prevents a clear separation of balloon induced turbulence from atmospheric turbulence by its spectral shape.

Accordingly, we do not have the possibility to independently evaluate the abundance of balloon-wake related turbulence in the wake detection algorithm by comparing with the LITOS data. Nevertheless, we found many instances of agreement between the wake prediction algorithm and LITOS measurements of balloon-wake created turbulence. One of them is shown in Figure 3. For an evaluation, more launches with an improved signal to noise ratio using a LITOS version that is not affected by wake from the payload chain would be desirable.

However, we took detailed care to include all possible effects into the uncertainty assessment. This makes us confident that the calculated probability for wake encounter gives a sound estimate of the real situation. This does not hold under turbulent conditions though, because wind shears on vertical scales below $200\,\mathrm{m}$ (e.g. Barat and Bertin, 1984b) will occur. In a radiosonde measurement, these scales cannot be unambiguously resolved due to mixture with instrumental effects (pendulum

motions of the gondola, bobbing motions of the balloon). Furthermore, the eddies created by the balloon will be advected by the larger eddies of the atmospheric turbulence, preventing a calculation of their path. Therefore, it is not possible to assess the payload-wake distance $d_{\text{p-wake}}$ under turbulent conditions.

According to Barat et al. (1984), $d_{\text{p-wake}}$ scales with the square of the balloon-gondola distance. This strongly increases the
likelihood of wake encounter for smaller payload-balloon distances. They have stated that 90 % of the data for a 2 m balloon at 100 m distance will be wake-free. From our analysis, we find that for this balloon-gondola distance of 100 m, only 27 % of all data points can be considered certainly wake-free. This value is considerably lower than the one acquired by Barat et al., even though we used a slightly smaller balloon. In contrast to their analysis we considered the measurement uncertainty in our calculation of the payload-wake distance and replaced their heuristic criterion for the minimal payload-wake distance by a
consideration of the transversal shape of the wake (Riddhagni et al., 1971; Gibson and Lin, 1968). Especially important is the consideration of the self-induced balloon motions as introduced by Scoggins (1965). They substantially increase the level of uncertainty in the payload-wake distance but have not been considered by Barat et al. (1984). This is crucial, because for the largest part of a standard radiosonde ascent, the balloon will be in the critical and supercritical Reynolds number range, where these motions occur. Accordingly, we assume that our lower number of certainly wake free altitude bins is explained by the
uncertainty in the wake prediction. This uncertainty is largely governed by effects inherent to sounding balloons (self-induced motions of the balloon, pendulum motions of the payload) that cannot be improved by advanced sensors.

From an analysis examining the influence of different parameters on the probability for wake encounter (Section 3.2.2) we find a strong dependence on wind shear and weaker dependencies on shears in the wind direction and on the relative velocity of the balloon. We like to stress that even though the former two cannot be influenced by the operator of the balloon, one
can reduce the probability for wake encounter by reducing the ascent speed of the balloon. This will furthermore reduce the amplitude of the self-induced balloon motions.

For the LITOS system as flown on 29 Jan 2016 (balloon-gondola distance of 180 m, balloon diameter up to 13 m), we expect 69 % of the flight to be wake free. The average probability for wake encounter is 5.6 %. Earlier measurements of our group (Theuerkauf, 2012; Haack et al., 2014; Schneider et al., 2015) were conducted with a smaller balloon-gondola distance (50 m)
and a larger balloon (up to 28 m) showing average wake encounter probabilities of about 60 %. Therefore, their geophysical results become questionable because of these wake related issues. Others (Schneider et al., 2017) already incorporated a precursor version of the payload-wake distance calculation that only lacked the uncertainty propagation presented here. Measurements with a critically low payload-wake distance have not been used in the latter publication, which is therefore considered to be sound within the limitations mentioned here. In order to completely avoid any wake influence we follow a technique proposed
by Kräuchi et al. (2016) on all newer LITOS measurements. It features two balloons with one of them being cut away at the highest point of the flight and the other one leading to a smooth downleg with a nearly constant descent rate.

Among different balloon-borne measurements, we expect wake related turbulence to have the strongest effects on high-resolution soundings of turbulent velocity fluctuations. This is, because the wake diameter will be between one and two balloon-diameters (Constantinescu and Squires, 2004; Riddhagni et al., 1971). Therefore, the strongest turbulent motions created by
a sounding balloon will be on scales below 2 m (ground level) and 26 m (top altitude, the precise value will depend on the

balloon type). On the other hand, larger scale distortions occur due to the radiosonde swinging in and out of the balloon's wake (Kräuchi et al., 2016), creating another temperature signal with a period of $\sim 5.5\,\mathrm{s}$ during nighttime and an additional signal with a period of $\sim 11\,\mathrm{s}$ during daytime for a radiosonde-balloon distance of $30\,\mathrm{m}$ (Tiefenau and Gebbeken, 1989). These scales are well resolved by the sampling rate of $1\,\mathrm{Hz}$ used on standard radiosondes. Therefore, Thorpe analysis results from these

measurements are likely to be affected by the balloon's wake. Further experimental studies on this topic seem desirable to us. An influential wake effect on high-resolution Thorpe analysis studies can be expected (e.g. Gavrilov et al., 2005; Luce et al., 2001, spatial resolution: $10\,\mathrm{cm}$, balloon-gondola distance: $100\,\mathrm{m}$). Results underlining this statement come from Wilson et al. (2011). They investigate the Thorpe-analysis of standard and high-resolution radiosondes, by identifying which inversions in the measured temperature profile are caused by true atmospheric overturnings. They find that in the troposphere this is the

case for only $11.4\,\%$ ($25\,\%$ for high-resolution data) of all inversions. They expect the remaining inversions to be strongly influenced by instrumental noise. From our work however, we expect small scale fluctuations due to the balloon's wake to have significantly contributed to their noise estimate. This expectation is supported by Jumper and Murphy (2001) finding a considerably higher amount of small amplitude spikes in the tropospheric temperature data of the ascent compared to the wake free descent.

In addition to these temperature related effects, Kräuchi et al. (2016) report humidity measurements from standard radiosondes to be affected as well, because the balloon's skin collects moisture in the troposphere, which is subsequently released in the stratosphere leading to a patchy contamination of the measurements while the radiosonde moves in and out of the balloon's wake. Gaffen (1994) points out that for long term temperature datasets caution is required, because an increase of the balloon-radiosonde distance on newer models decreases the effect of the daytime heating of the balloon's wake, thereby lead-

ing to false cooling trends on daytime data. Luers and Eskridge (1998) note that solar radiation is a stronger concern for very short payload-balloon distances especially in the stratosphere, as the convective cooling of the temperature sensor is decreased because of the reduced relative velocity $w_{\mathrm{res}}$ in the wake.

Azouit and Vernin (2005) expect the wake not to have any significant effect on their measurements of the refractive index structure function constant $C_N^2$, which is related to atmospheric turbulence. Applying the Barat et al. (1984) criterion they find

less than $2\,\%$ of all altitude bins to be affected. However, they report a mean wind shear three times as high as the one from the radiosonde dataset used here, which can explain their lower number of altitude bins with an expected balloon-wake influence.

Concerning wake influences from smaller objects in the vicinity of the sensor we notice that they can usually be identified by their spectral shape. However, if not removed from the measurement they cause false detections of strong turbulence in the order of $100\,\mathrm{mW\,kg^{-1}}$. This may be taken as a reminder that even small objects of millimetre size cause severe turbulent

fluctuations up to metre-scales for at least 150 object diameters downstream of the flow disturbance.

Answering our question from the introduction: Yes, we can determine regions in the dataset that are prone to balloon-wake related measurement distortions by an automated calculation of the likelihood for balloon-wake encounter, even though there is a considerable uncertainty in the computation. Regions influenced by the wake from smaller objects in the payload chain can be identified only manually by their spectral shape of the high-resolution data.

# 6  Conclusions

In this article we have classified two major distortions of turbulence measurements that occur on rising balloons: Turbulence caused by the balloon and turbulence caused by small objects in the payload chain. For the former, we developed a tool to compute the likelihood of wake encounter at the payload position. This is done by calculating the drift of the balloon's wake using radiosonde data and applying a full uncertainty analysis. However, such an assessment is generally not possible under turbulent conditions due to the advection of the balloon's wake. The uncertainty has been reduced by adapting the vertical wind retrieval from Wang et al. (2009) to our larger balloons and to the different launch preparation procedures. Furthermore, we developed a statistical approach to approximate the size of the wake that is based on laboratory studies. We found instances of good agreement between our balloon-wake prediction and the LITOS data. The abundance of these wake encounters however cannot be unambiguously evaluated, because the balloon's wake is masked for about one third of the flight due to turbulence created by small objects. The latter was found to be not accessible to an automatic detection. Instead, we sort those areas out by manual inspection of the spectral shape of all turbulent altitude bins.

By analysing a set of 30 radiosondes with our wake assessment tool we found that for a standard radiosonde configuration only about 4 % of the flight can be certainly considered wake free, with an average wake probability of about 28 %. The low abundance of certainly wake free regions also reflects the uncertainty in the wake assessment, which is largely caused by motions inherent to sounding balloons. From a general perspective, measurements resolving scales in the centimetre range or below will be additionally susceptible to the wake of small objects like our ropes. Our study shows however that the wake of large objects like the balloon will influence measurements of standard radiosondes with meter scale resolution as well. Deduction of atmospheric turbulence parameters from radiosonde balloons can be seriously flawed if wake effects are ignored. This calls for thorough investigations of wake effects on sounding balloon measurements and for even longer payload-balloon distances than the 55 m currently used on standard radiosondes. For research purposes where the complete avoidance of any wake influence is crucial (e.g. turbulence measurements, high accuracy temperature soundings), we strongly recommend to measure on a descending balloon with the sensor pointing downward.

# 7  Data and source code availability

The MATLAB® source code for the wake evaluation tool (Section 3.1) and for the vertical wind retrieval (Appendix A1) is published online at ftp://ftp.iap-kborn.de/data-in-publications/SoederAMT2019/.

Besides the source code, this repository contains a user guide and the data underlying Figure 4 as a running example. The radiosonde data used in Section 3.2.1 can be obtained via the HALO database (https://halo-db.pa.op.dlr.de/mission/3). The LITOS data and more specialised source code will be made available on request to the corresponding author.

**Appendix A: Vertical wind retrieval from radiosonde data**

**A1   Calculation of vertical winds**

For the retrieval of energy dissipation rates from the LITOS instrument and for our wake prediction algorithm we need to obtain the relative vertical velocity $w_{\mathrm{rel}}$ between the balloon and the sonde. Since LITOS does not measure vertical winds, we retrieve them from the ascent rate of the radiosonde instead. The variations of ascent rate however, are not directly proportional to the vertical wind, because of changing flow conditions around the balloon (e.g. Gallice et al., 2011). We follow the approach of Wang et al. (2009), who obtained vertical winds by using a parametrisation of the balloon ascent based on the balance of the drag force and the buoyancy force of the balloon. We measure the lift of our balloon during the filling with an uncertainty of $\pm 5\,\mathrm{N}$. In the data post processing, we optimise the value for the balloon lift at launch and the drag coefficient of the balloon such that the median of the retrieved vertical wind is minimal (criterion based on Wang et al., 2009). For easier comparison, our article adopts the notation of Wang et al. (2009).

In more detail, we write the buoyancy force of the balloon BF as the difference of the lifting force and the accumulated weight force due to the masses of the payload ($m_{\mathrm{p}}$), the balloon ($m_{\mathrm{b}}$) and the lifting gas (helium, $m_{\mathrm{He}}$):

$$\mathrm{BF} = g\mathrm{BV}\rho - g(m_{\mathrm{p}} + m_{\mathrm{b}} + m_{\mathrm{He}}), \tag{A1}$$

where $g$ denotes the gravitational constant, $\rho$ the air density and BV the balloon volume, which can be expressed using the balloon volume at launch $\mathrm{BV}_0$ and the air density at launch $\rho_0$ by $\mathrm{BV} = \mathrm{BV}_0\rho_0/\rho$. Furthermore, the helium mass can be expressed in terms of the helium density on ground level ($\rho_{\mathrm{He}}$) by $m_{\mathrm{He}} = \mathrm{BV}_0\rho_{\mathrm{He}}$. Therewith, the balloon volume at launch is written as:

$$\mathrm{BV}_0 = \frac{\mathrm{BF}_0/g + m_{\mathrm{b}}}{\rho_0 - \rho_{\mathrm{He}}}, \tag{A2}$$

$\mathrm{BF}_0$ is the lifting force of the balloon at launch. We measure the lifting force of our balloon during the filling process $\mathrm{BF}_{\mathrm{f}}$ inside a hangar with an uncertainty of $\pm 5\,\mathrm{N}$. Inside the hangar, we find a different temperature $T_{\mathrm{f}}$ compared to the outside temperature at launch $T_0$. This leads to a non-adiabatic loss in balloon volume and lifting force during the subsequent launch preparations outside the hangar. Assuming ideal gas law ($\mathrm{BV}_0 = \mathrm{BV}_{\mathrm{f}}\frac{T_0}{T_{\mathrm{f}}}$), we rewrite Equation A1 for the buoyancy force of the balloon using Equation A2:

$$\mathrm{BF} = (gm_b + \mathrm{BF}_{\mathrm{f}})\frac{T_0}{T_{\mathrm{f}}} - g(m_{\mathrm{p}} + m_{\mathrm{b}}), \tag{A3}$$

As mentioned above, during flight the buoyancy force BF equals the drag force DF of the balloon. The latter is given by:

$$\mathrm{DF} = (c_{\mathrm{db}}A_{\mathrm{b}} + c_{\mathrm{dp}}A_{\mathrm{p}})\rho w_{\mathrm{rel}}^2/2. \tag{A4}$$

$c_{\mathrm{db}}$ denotes the drag coefficient of the balloon, $A_{\mathrm{b}}$ its cross-sectional area and $w_{\mathrm{rel}}$ the relative vertical velocity between the balloon and the surrounding air. $c_{\mathrm{dp}}$ stands for the drag coefficient of the payload and $A_{\mathrm{p}} = 0.5\,\mathrm{m}^2$ for the accumulated cross-sectional area of all payload boxes in the case of LITOS. The shapes of the payload boxes include cuboids, spheres and cones.

We assume their drag coefficient to be $c_{dp} = 1$ on average. Accordingly, balancing the buoyancy force and the drag force of the balloon and using Equation 2 yields for the vertical wind:

$$w = w_{asc} - \sqrt{2BF/((c_{db}A_b + c_{dp}A_p)\rho)} \tag{A5}$$

From this equation, in combination with Equation A3, $c_{db}$ and $BF_f$ are fitted so that the median of all retrieved vertical winds
$w$ over the whole flight is minimised (approach similar to Wang et al., 2009). The drag coefficient $c_{db}$ however, depends on the flow conditions around the balloon. These flow conditions are characterised by the Reynolds number:

$$Re = \frac{w_{rel}D_{bal}}{\nu}, \tag{A6}$$

$w_{rel}$ is the relative vertical velocity between the balloon and the atmosphere, $D_{bal}$ is the balloon diameter and $\nu$ the kinematic viscosity according to Equation 4. As $w_{rel}$ is a result of the vertical wind retrieval, we need to make an initial guess for $Re$ by
assuming that $w = 0$ and therefore $w_{rel} = w_{asc}$. In a second run, $Re$ is calculated as described in Equation A6 and shown in the left panel of Figure A1 using a cut-off period of 40 s to remove bobbing motions of the balloon (caused by the flexibility of the balloon material), self-induced motions of the balloon and pendulum motions of the payload. For the initial guess, all changes in ascent rate due to changing aerodynamic forces on the balloon are removed by a digital low pass filter with a cut-off period of 400 s.

The resulting ascent rate in still air is shown by the black line in the right panel of Figure A1. The resulting subcritical drag coefficient is $c_{db} = 0.56$, which is slightly higher than the 0.50 to 0.51 obtained by Achenbach (1974) for smooth and marginally roughened spheres. The resulting lifting force is 75 N. The remaining median of the vertical wind (fit residuum) is $2 \cdot 10^{-10}\,\mathrm{m\,s^{-1}}$. Wang et al. (2009) found a mean fit residuum of $0.02\,\mathrm{m\,s^{-1}}$ for 102 radiosondes launched during the Terrain-induced Rotor Experiment (T-Rex) in 2006. This shows a sufficient fit quality in our adapted version of the model.
For comparison of our retrieved vertical winds we use the Weather Research and Forecasting (WRF) model with a horizontal resolution of 800 m (setup similar to Schneider et al. (2017)). We compare the vertical from our retrieval algorithm with vertical winds winds from the model along the flightpath of the radiosonde. In the critical and supercritical Reynolds number range (below $\approx 19\,\mathrm{km}$ altitude) this model shows vertical winds of up to $\pm 2\,\mathrm{m\,s^{-1}}$ along the flight track (not shown here), while our retrieval assumes $w = 0$. Therefore, the error of setting $w = 0$ is below $\pm 2\,\mathrm{m\,s^{-1}}$ in our case. In the subcritical Reynolds
number range (above $\approx 19\,\mathrm{km}$ altitude) the deviation between retrieved vertical winds and model data is below $1\,\mathrm{m\,s^{-1}}$ for altitudes below 26 km. Above, the amplitudes of the model decrease sharply, possibly caused by the damping layer of the model starting at 30 km altitude. As a rough guess, we may estimate the error in the vertical wind above 19.5 km to be below $\pm 1\,\mathrm{m\,s^{-1}}$.

   We are aware that there are effects influencing the ascent rate that are not included in the model. Presumably, the most
important one is a temperature difference of the lifting gas to the surrounding air. Gallice et al. (2011) developed an extensive model considering the temperature distribution of the lifting gas inside the balloon. However, according to the authors it is applicable to night time launches only and needs an experimentally acquired drag curve for the particular balloon type. Both conditions make their calculations unsuitable to our dataset.

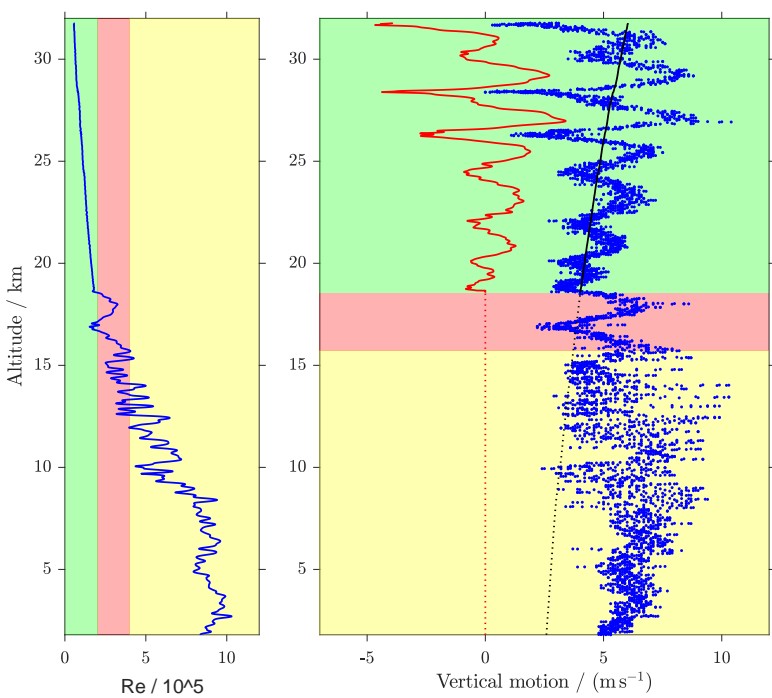

**Figure A1.** Left: Reynolds number of the flow around the balloon according to Equation A6. Right: unfiltered ascent rates of the balloon (blue dots). Relative vertical velocity of the balloon $w_{\mathrm{rel}}$ (black), vertical winds according to Equation A5 (red). Vertical winds in the supercritical and critical Reynolds number regime are set to zero because of changing drag coefficients. Retrieved relative vertical velocities underestimate the ascent rates in these regimes and are shown as a dashed black line. Both panels: Subcritical (green), critical (red) and supercritical (yellow) flow regime.

*Author contributions.* JS acquired LITOS evaluated the LITOS data, prepared the software and wrote the manuscript with feedback from the other authors. MG supervised the acquisition of the LITOS data and contributed to the discussion of the manuscript. AS provided parts of the software that is used to retrieve energy dissipation rates. AD supported the discussion of the manuscript and supervised the GW-LCycle measurement campaign under which the radiosonde data series and part of the LITOS data used here have been acquired. HW contributed to the discussion of the manuscript and the uncertainty estimate in the wake evaluation tool in particular. JW set up the WRF simulations that are used as a comparison for the vertical wind retrieval. FJL took part in the discussion of the manuscript and provided the general concept behind the LITOS measurement.

*Competing interests.* The authors declare that they have no conflict of interest.

*Acknowledgements.* We like to thank Christoph Zülicke and Mark Schlutow for countless insightful discussion on the subject of this publication. Moreover, we acknowledge the help of Reik Osterman and Michael Priester with the development of the LITOS instrument.

We greatly appreciate the funding received from the Bundesministerium für Bildung und Forschung (BMBF, Federal Ministry of Education and Research) under project 01 LG 1218A (ROMIC-METROSI). The radiosonde data used in this study have been acquired during the GW-LCycle II campaign of the project 01 LG 1206A (ROMIC-GWLcycle). Many fruitful discussions on the subject were made possible in the framework of the MS-GWaves project by the Deutsche Forschungsgemeinschaft (DFG, German Research Foundation). Last but not least we like to thank Richard Wilson and one anonymous referee for their comments and Jörg Gumbel for handling the manuscript.

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
