# Peer review of "Evaluation of wake influence on high-resolution balloon-sonde measurements"

_Atmospheric Measurement Techniques, 2018_

## Referee Comment (RC1) · Richard Wilson (Referee) · 14 Mar 2019

Review report on the paper of Söder et al. for the Journal Atmospheric Measurements Techniques: **Evaluation of wake influence on high-resolution balloon-sonde measurements**.

**General comments**

The article "**Evaluation of wake influence on high-resolution balloon-sonde measurements**" addresses the possible impact of the wake-created fluctuations on turbulence measurements from rising balloons. Such a wake can be generated in the trail of the balloon and/or by objects in the vicinity of the sensors (gondola, rope). The authors show few examples of likely wake encounter based on very high-resolution (8 kHz) measurements of the wind velocity from their LITOS instrument. These examples are convincing as they show large velocity fluctuations hardly distinguishable from atmospheric turbulence when the payload is likely in the balloon wake.

The approach of this work is mostly probabilistic, the main aim is to estimate the probability for wake encounter by the payload hanged below a rising balloon. Based on a probabilistic model, the authors estimate the relative impact of various factors, some of them not being considered in previous works such as the vertical wind or the rotation of the wind vector. As expected, the probability for the payload to be in the wake depends mainly on the balloon-gondola distance and on the vertical shear of the horizontal wind. From a statistical study based on 30 radiosondes, they estimated the probability for the payload to be in the wake of the balloon. They concluded that the probability for such an encounter for standard radiosondes is 28% in the average.

The article addresses a relevant question: to what extent does the disturbances induced by the system carrying a high resolution sensor impact turbulence measurement? The authors convincingly show (1) some effects of such disturbances (2) that the probability for the payload to be in the wake is quite large even for large balloon-payload distances, and (3) that this issue must be carefully addressed when estimating atmospheric turbulence from instruments carried by rising balloons. The possible wake effects are obvious for high-resolution sensors, but may also impact turbulence detections from standard resolution (~1 Hz) radiosondes.

Undoubtedly, this article deserves to be published. It is well organized, well written (as far as I can judge). The figures are appropriate and well made. The quotes seem relevant to me. I generally appreciated this work.

**Specific comments**

1) I wonder about the possible impact of wake on the turbulence detection from standard radiosondes. This issue could be addressed by considering the statistics of the time during which the payload stays in the wake, i.e. of the spatial extent of the payload-wake encounters. The two presented example have spatial extent of 15 and 6 m, hardly detectable from radiosonde measurements (some authors - Ferron et al., Wilson et al. - recommend to undersample the vertical profile in order to detect inversions in the potential temperature profile. For radiosondes, this lead to vertical resolution of about 15 m). Do the authors think that such a statistics could be obtained from the presented probabilistic model? Perhaps beyond the subject of the paper, I think such a result could increase the scope of this work.

2) p17, l5-6: Can you be more specific about this affirmation?

3) p21, l15-16: the assertion that the wake of the balloon contributes to noise (meaning instrumental noise) is questionable. The signatures of the balloon's wake on the temperature profile, either temperature peaks or turbulent eddies, are not a contribution to instrumental noise (assumed uncorrelated), but are likely responsible of false inversions in the potential temperature profile.

**Minor comments**

p8, l13: Euklidian -> Euclidian

p 8, l10: why a factor 2 in the definition of L?

p8, l10: the notation $L = (...).\min(w\_rel)$ is not very satisfactory (the dot can be read as an operator...)

Appendix and figure A1: Can one conclude that $w$ is estimated to be zero in the troposphere for all flights?

---

## Referee Comment (RC2) · Anonymous Referee #2 · 29 Apr 2019

This manuscript provides a comprehensive analysis of wake effects on balloon-borne measurements. The analysis is focused on highly sophisticated turbulence measurements of the LITOS project but provides also important information for other balloon experiments. The analysis is sound, and the manuscript is well written and very suited for publication in AMT. I recommend publication after a minor revision. My comments mainly concern clarifications that probably would make the paper more accessible/useful to a broader audience.

What is actually a typical wake effect on measurements of turbulence (or other quantities like temperature or humidity)? Please provide more information about this to the unexperienced reader. The manuscript seems to provide examples both of increased

turbulence and decreased turbulence in the wake (figure 3; page 11, line 5-9; page 11, line 12-13).

At several places in the manuscript it is stated that the wake does not have a clear outer boundary. Rather, one would expect, a continuous transition from a perturbed region to an unperturbed region. On the other hand you use the notation "inside the wake" and "outside the wake" throughout the manuscript. And you introduce e.g. the "radius of the wake" (page 10, line 11). This is not consistent and should be clarified.

Most analysis in the paper is done in terms of likelihood for instruments to encounter the wake of the balloon (or of the payload chain). What would desirable for balloon researchers is a more "deterministic" algorithm that provides a rather clear statement Yes or No about being influenced by a wake effects in a given situation. Do the authors see any way forward towards developing such a "deterministic" analysis tool?

On page 11, line 12-13, you refer to probabilities exceeding 95% as "wake affected within uncertainty". Would you not consider significantly lower probabilities (e.g. 80% or 60%) as being "wake affected within uncertainty". What is the significance of the number 95%?

Related to the above questions: On page 4, line 18, you refer to a measurement unperturbed by ant wake effect. How do you know this?

It would be instructive if you in your conclusions formulated some general advice for balloon researcher about how to deal with wake effects.

Some minor comment:

Om page 3, lines 9, 12, 15: It is unclear what "their" refers to in this paragraph. It somehow refers to "other studies" in line 8. Please clarify the formulations.

On page 9, line 9-12, you introduce the amplitude of the balloons horizontal motion. It would be instructive to provide the reader with some typical numbers for this amplitude, e.g. for the LITOS case.

---

## Author Response (AR1)

**1 Response to reviewer comments**

In the following, we provide a point-by-point reply to both reviews. The text from the reviews is highlighted by italics.

**1.1 Review #1**

We like to thank Richard Wilson for reviewing our manuscript and express our gratitude for his suggestions. A point-by-point answer to his comments is given in the following.

*The article "Evaluation of wake influence on high-resolution balloon-sonde measurements" addresses the possible impact of the wake-created fluctuations on turbulence measurements from rising balloons. Such a wake can be generated in the trail of the balloon and/or by objects in the vicinity of the sensors (gondola, rope). The authors show few examples of likely wake encounter based on very high-resolution (8 kHz) measurements of the wind velocity from their LITOS instrument. These examples are convincing as they show large velocity fluctuations hardly distinguishable from atmospheric turbulence when the payload is likely in the balloon wake. The approach of this work is mostly probabilistic, the main aim is to estimate the probability for wake encounter by the payload hanged below a rising balloon. Based on a probabilistic model, the authors estimate the relative impact of various factors, some of them not being considered in previous works such as the vertical wind or the rotation of the wind vector. As expected, the probability for the payload to be in the wake depends mainly on the balloon-gondola distance and on the vertical shear of the horizontal wind. From a statistical study based on 30 radiosondes, they estimated the probability for the payload to be in the wake of the balloon. They concluded that the probability for such an encounter for standard radiosondes is 28% in the average. The article addresses a relevant question: to what extent does the disturbances induced by the system carrying a high resolution sensor impact turbulence measurement? The authors convincingly show (1) some effects of such disturbances (2) that the probability for the payload to be in the wake is quite large even for large balloon-payload distances, and (3) that this issue must be carefully addressed when estimating atmospheric turbulence from instruments carried by rising balloons. The possible wake effects are obvious for high-resolution sensors, but may also impact turbulence detections from standard resolution ( 1 Hz) radiosondes.*

*Undoubtedly, this article deserves to be published. It is well organized, well written (as far as I can judge). The figures are appropriate and well made. The quotes seem relevant to me. I generally appreciated this work.*

We are grateful for this kind assessment of our article.

*1) I wonder about the possible impact of wake on the turbulence detection from standard radiosondes. This issue could be addressed by considering the statistics of the time during which the payload stays in the wake, i.e. of the spatial extent of the payload-wake encounters. The two presented example have spatial extent of 15 m and 6 m, hardly detectable from radiosonde measurements (some authors - Ferron et al., Wilson et al. - recommend to undersample the vertical profile in order to detect inversions in the potential temperature profile. For radiosondes, this lead to vertical resolution of about 15 m). Do the authors think that such a statistics could be obtained from the presented probabilistic model? Perhaps beyond the subject of the paper, I think such a result could increase the scope of this work.*

We clearly share the opinion that retrieving the proposed statistics would increase the scope of this work. Unfortunately however, we do not see a way forward to obtain the necessary information. This is the case for two reasons:

- Retrieving the statistics from our LITOS measurement does not work, because the wake from the ropes partly masks the wake from the balloon thereby introducing a bias towards smaller spatial extends of the payload-wake encounters.

- Retrieving the statistics purely from our wake detection algorithm might be questionable, because the wake encounter probability does not give a clear discrimination between wake affected and wake free altitude bins. We chose to discard all turbulence measurements from altitude bins showing $P_{\text{wake}} > 5\,\%$ in order to avoid wake influence on our turbulence measurements. Retrieving such statistics using the $5\,\%$ threshold however, would overestimate the spatial extend of the payload-wake encounters.

Nevertheless, we still assume that these wake encounters may influence Thorpe analyses from standard radiosondes even if the dataset is undersampled to a vertical resolution of about 15 m. Tiefenau and Gebbeken (1989) find periods of 5.5 s and 11 s on radiosonde temperature data caused by the payload swinging in and out of the balloon's wake. We assume that such long periods may be detectable even in undersampled radiosonde data sets.

*2) p17, l5-6: Can you be more specific about this affirmation?*

In order to solidify this affirmation, we added a visualization of the ninety-fifth percentile of the radiosonde dataset to the plots of this section. Furthermore, we calculated the spread of the wake probability depending on changes in wind shear, rotation of the wind vector and changes in relative vertical velocity, respectively. The revised text reads: "From the analysis of these three parameters, we find that within the ninety-fifth percentile of the wind shear for the given radiosonde dataset, the wake probability changes from $0.1\,\%$ to $96\,\%$. Within the ninety-fifth percentile of all examined rotations in the horizontal wind vector the wake probability changes from $68\,\%$ to $95\,\%$. For the variation in relative vertical balloon velocity the wake probability changes from $43\,\%$ to $84\,\%$. Therefore, we conclude that within the spread of the given radiosonde dataset wind shear has the strongest influence on the likelihood for wake encounter."

*3) p21, l15-16: the assertion that the wake of the balloon contributes to noise (meaning instrumental noise) is questionable. The signatures of the balloon's wake on the temperature profile, either temperature peaks or turbulent eddies, are not a contribution to instrumental noise (assumed uncorrelated), but are likely responsible of false inversions in the potential temperature profile.*

We agree with the reviewer that our assumption resulted from misinterpreting the role of instrumental noise in Wilson et al. (2010) and Wilson et al. (2011). The corresponding statements have been deleted from our discussion.

Minor comments

*p8, l13: Euklidian -> Euclidian*

Thanks for pointing out this typo. The correction has been made.

*p 8, l10: why a factor 2 in the definition of L?*

The factor of two in the definition of L is a safety margin. It is needed, because in case the wind shear changes over the distance between the payload and the balloon the

wake will not move along a straight line and will therefore need more timesteps to reach the closest distance to the payload. However, we agree that a factor of 2 is excessive for this purpose. In order to save computational cost, this factor has been reduced to 1.2 in the revised manuscript.

*p8, l10: the notation L = (...). min(w\_rel) is not very satisfactory (the dot can be read as an operator...)*

Thanks for the suggestion. We changed the sentence to " L = (...). In this case, min(w\_rel)".

*Appendix and figure A1: Can one conclude that w is estimated to be zero in the troposphere for all flights?*

We cannot sensibly estimate $w$ in the troposphere due to aerodynamically induced variations in the ascent rate. Our wake prediction algorithm however needs an input for $w$. Therefore, we set $w = 0$ in the critical and supercritical Reynolds number range. To make this more visible, we replaced the solid line in Figure A1 by a dotted line where $w$ is set to zero.

**1.2 Review #2**

We are grateful for the effort undertaken by the reviewer and like to thank her/him for the constructive recommendations. A point-by-point answer is given below.

*This manuscript provides a comprehensive analysis of wake effects on balloon-borne measurements. The analysis is focused on highly sophisticated turbulence measurements of the LITOS project but provides also important information for other balloon experiments. The analysis is sound, and the manuscript is well written and very suited for publication in AMT. I recommend publication after a minor revision. My comments mainly concern clarifications that probably would make the paper more accessible/useful to a broader audience.*

We are thankful for the positive feedback from the reviewer. We hope that our changes will increase the accessibility of our manuscript.

*What is actually a typical wake effect on measurements of turbulence (or other quantities like temperature or humidity)? Please provide more information about this to the unexperienced reader.*

Thanks very much for your suggestion. We have added the following paragraph to the introduction that highlights the impact of wake created turbulence.

"On measurements of turbulent velocity fluctuations, the wake from the balloon can hardly be distinguished from atmospheric turbulence of the same strength. With our LITOS instrument we found that the spectral shape of the velocity fluctuations does not allow a distinction between atmospheric turbulence and wake. Depending on the payload-balloon distance, we found dissipation rates created by the balloon's wake between $10^{-4}\,W\,kg^{-1}$ and $10^{-2}\,W\,kg^{-1}$. In terms of aviation turbulence categories, these dissipation rates correspond to "light" and "moderate" turbulence using the scaling of Sharman et al. (2014) for medium-sized aircraft. Wake effects from the ropes holding the gondola show consistently "severe" turbulence intensities around $10^{-1}\,W\,kg^{-1}$. Accordingly, these effects should not be neglected for turbulence measurements from rising balloons. For standard radiosondes Kräuchi et al. (2016) report a warm bias of $1\,K$ on average for a daytime sounding in the stratosphere. Furthermore, moisture from the balloon's skin will lead to a wet bias of stratospheric humidity if the sensor is in the balloon's wake."

*The manuscript seems to provide examples both of increased turbulence and decreased turbulence in the wake (figure 3; page 11, line 5-9; page 11, line 12-13).*

According to our understanding, the strength of wake related turbulence depends on the payload-balloon distance and possibly on the size of the balloon or the ascent rate. We found kinetic energy dissipation rates from $10^{-4}\,W\,kg^{-1}$ to $10^{-2}\,W\,kg^{-1}$. This, however, is not related to the strength of atmospheric turbulence. Therefore, atmospheric turbulence may be weaker as well as stronger than wake related turbulence. In order to point this out, we changed the paragraph following page 11, line 7:

"We like to stress, however, that the data underlying Figure 1 show only one exemplary case of a turbulent altitude bin. Under different atmospheric conditions, LITOS measured atmospheric turbulence ranging from $0.001\,mW\,kg^{-1}$ to $100\,mW\,kg^{-1}$. In contrast, we typically find dissipation rates between $0.01\,mW\,kg^{-1}$ and $1\,mW\,kg^{-1}$ for balloon-wake induced turbulence. However, during previous measurements with lower payload-balloon distances we measured wake induced turbulence stronger than $10\,mW\,kg^{-1}$ (data not

shown here)."

*At several places in the manuscript it is stated that the wake does not have a clear outer boundary. Rather, one would expect, a continuous transition from a perturbed region to an unperturbed region. On the other hand you use the notation "inside the wake" and "outside the wake" throughout the manuscript. And you introduce e.g. the "radius of the wake" (page 10, line 11). This is not consistent and should be clarified.*

We are sorry for our inconsistent terminology on this matter. To our understanding, there is a clear outer boundary of the wake. However, the transversal distance of the outer boundary of the wake to the wake's centre will change depending on the downstream distance from the balloon. This is, because the outer boundary of the wake is shaped by larger eddies. A visualisation of the flow in a turbulent wake showing a fairly clear outer boundary is given in Jang and Lee (2008, Figure 11b). Even though we expect a clear outer boundary, we cannot state its distance from the wake centre for a given downstream distance to the balloon due to the chaotic flow in the wake. This is one of the main reasons to take a probabilistic approach in our wake detection algorithm. The sentence on page 3, lines 3-7 has been changed to:

"Furthermore, we consider that the diameter of the balloon's wake changes on short time scales of a few seconds due to the production of larger vortices. Furthermore, its mean diameter increases on longer timescales in the order of several 10 seconds. Since the balloon's contour resembles a sphere during flight, we can refer to fundamental experiments done in wind tunnels (e.g. Riddhagni et al., 1971; Gibson and Lin, 1968). An informative visualisation of such a flow can be found in Jang and Lee (2008, Figure 11)."

Furthermore, page 7, line 5 has been amended. It reads:

"Due to the intermittent nature of the turbulent wake, its diameter changes on downstream length scales smaller than a few balloon diameters."

*Most analysis in the paper is done in terms of likelihood for instruments to encounter the wake of the balloon (or of the payload chain). What would desirable for balloon researchers is a more "deterministic" algorithm that provides a rather clear statement Yes or No about being influenced by a wake effects in a given situation. Do the authors see any way forward towards developing such a "deterministic" analysis tool?*

We fully agree with the reviewer that a deterministic tool to determine whether the balloon's gondola is inside or outside the wake would be highly desirable. Unfortunately however, we do not see a way to create such a tool using radiosonde wind data. The main reason for this conclusion is that the uncertainty in the payload-wake distance ($\Delta_{d_{\mathrm{p-wake}}}$) is in the same order of magnitude as the payload-wake distance ($d_{\mathrm{p-wake}}$). From our point of view this requires an uncertainty analysis, which is done in our probabilistic approach. The uncertainty in the position of the balloon ($\Delta_{X_{\mathrm{p-wake}}}$) could only be reduced by using an additional GPS receiver attached to the balloon. The uncertainties in the radiosonde wind measurement ($\Delta_U$) and in the payload position ($\Delta_{X_{\mathrm{p}}}$) however, could be generally reduced by enhancing precision and sampling rate of the radiosonde. Nevertheless, these measures would still not remove the uncertainty in the diameter of the wake (as discussed in the previous point). In conclusion, one could reduce the uncertainties in the calculation with a high technical effort, but would still not be able to fully avoid a probabilistic approach. Therefore, we assume that our approach provides a

helpful compromise between usability and certainty in the prediction.

*On page 11, line 12-13, you refer to probabilities exceeding 95% as "wake affected within uncertainty". Would you not consider significantly lower probabilities (e.g. 80% or 60%) as being "wake affected within uncertainty". What is the significance of the number 95%?*

Our aim is to sort out all data bins that are not most certainly wake free. Therefore, we only use turbulence measurements from altitude bins showing $P_{wake} < 5\%$. From our point of view a significantly lower confidence level (accepting turbulence measurements with $P_{wake} \gg 5\%$ would illicitly increase the risk of misinterpreting wake created turbulence for atmospheric turbulence. Accordingly, we define altitude bins with a wake probability above 95% as wake affected in order to be consistent with our definition of wake free altitude bins. To make our wording more precise, we removed "within uncertainty" from page 11, line 13 of the manuscript.

*Related to the above questions: On page 4, line 18, you refer to a measurement unperturbed by ant wake effect. How do you know this?*

We know that there are no wake effects on this measurement, because we measured on a descending balloon with our sensors pointing downward. Accordingly, all parts of the payload are located downstream of our sensors. Therefore, the sensors cannot be hit by the wake from these objects.

*It would be instructive if you in your conclusions formulated some general advice for balloon researcher about how to deal with wake effects.*

Thank you very much for pointing this out. We added the following sentence to our conclusion:

"For research purposes where the complete avoidance of any wake influence is crucial (e.g. turbulence measurements, high accuracy temperature soundings), we strongly recommend to measure on a descending balloon with the sensor pointing downward."

*Some minor comment: Om page 3, lines 9, 12, 15: It is unclear what "their" refers to in this paragraph. It somehow refers to "other studies" in line 8. Please clarify the formulations.*

Thanks for identifying this issue. We changed the sentences to:

"The most common method to obtain energy dissipation rates from radiosonde temperature profile has been adapted from oceanic sciences by Luce et al. (2002) and Clayson and Kantha (2008). It is frequently referred to as the "Thorpe analysis." Energy dissipation rates are inferred from the vertical displacement (Thorpe displacement) of an air parcel compared to a statically stable profile (Wilson et al., 2010, 2011). Typically, for a standard radiosonde the distance between the balloon and the sensor is between 30 m and 55 m. This makes the measurements susceptible to distortions from the balloon's wake (e.g. Jumper and Murphy, 2001; Kräuchi et al., 2016). Hence, our wake evaluation tool may be used to assess the likelihood of wake influence for every altitude bin of a Thorpe analysis turbulence retrieval, depending on the balloon-payload distance of the instrument."

*On page 9, line 9-12, you introduce the amplitude of the balloons horizontal motion. It would be instructive to provide the reader with some typical numbers for this amplitude, e.g. for the LITOS case.*

We have added the following statement to the text: "Typically, $D_{bal}$ is below 10 m.

In the case of the LITOS launch from 29 January 2016 (discussed in Section 3.1.4) the mean amplitude in the critical and supercrititcal Reynolds number range is 6.3 m."

**2 Further changes to the manuscript**

In the following, we list all relevant changes to the manuscript that are not related to the reviews.

- p. 4, l .16:
  "The geometric mean of the dissipation rate from both sensors is $9.9\,\mathrm{mW\,kg^{-1}}$, which corresponds to a moderate turbulence intensity according to aviation standards (Sharman and Pearson, 2017)." $\rightarrow$ "The geometric mean of the dissipation rate from both sensors is $9.9\,\mathrm{mW\,kg^{-1}}$, which corresponds to a moderate turbulence intensity for medium-sized aircraft according to aviation standards (Sharman et al., 2014)."

- p.8, l. 15:
  $d_{\mathrm{p\text{-}wake}}(t^l) = |\vec{X}_p(t^l) - \vec{X}_{\mathrm{wake}}(t^l)| \; \rightarrow$
  $d_{\mathrm{p\text{-}wake}}(t^l) = \min_{m=0,...,L} |\vec{X}_p(t^l) - \vec{X}_{\mathrm{wake}}(t^l)|$

**3 Marked-up manuscript**

[revised manuscript text omitted]

30  Vaisala: Vaisala Radiosonde RS41-SG – accuracy and reliability, Vaisala Corporation, https://www.vaisala.com/sites/default/files/documents/RS41-SG-Datasheet-B211321EN.pdf, 2018.

Wang, J., Bian, J., Brown, W. O., Cole, H., Grubišić, V., and Young, K.: Vertical Air Motion from T-REX Radiosonde and Dropsonde Data, Journal of Atmospheric and Oceanic Technology, 26, 928–942, doi:10.1175/2008JTECHA1240.1, 2009.

35  Williamson, C. H.: Vortex dynamics in the cylinder wake, Annual Review of Fluid Mechanics, 28, 477–539, doi:10.1146/annurev.fl.28.010196.002401, 1996.

Wilson, R., Luce, H., Dalaudier, F., and Lefrère, J.: Turbulence Patch Identification in Potential Density or Temperature Profiles, Journal of Atmospheric and Oceanic Technology, 27, 977–993, doi:10.1175/2010JTECHA1357.1, 2010.

Wilson, R., Dalaudier, F., and Luce, H.: Can one detect small-scale turbulence from standard meteorological radiosondes?, Atmospheric Measurement Techniques, 4, 795–804, doi:10.5194/amt-4-795-2011, 2011.